# A critical role of PRDM14 in human primordial germ cell fate revealed by inducible degrons

Anastasiya Sybirna[1,2,3], Walfred W.C. Tang[1,2], Merrick Pierson Smela [1,2], Sabine Dietmann[3], Wolfram H. Gruhn[1,2], Ran Brosh [4] & M. Azim Surani [1,2,3✉]

PRDM14 is a crucial regulator of mouse primordial germ cells (mPGCs), epigenetic reprogramming and pluripotency, but its role in the evolutionarily divergent regulatory network of human PGCs (hPGCs) remains unclear. Besides, a previous knockdown study indicated that PRDM14 might be dispensable for human germ cell fate. Here, we decided to use inducible degrons for a more rapid and comprehensive PRDM14 depletion. We show that PRDM14 loss results in significantly reduced specification efficiency and an aberrant transcriptome of hPGC-like cells (hPGCLCs) obtained in vitro from human embryonic stem cells (hESCs). Chromatin immunoprecipitation and transcriptomic analyses suggest that PRDM14 cooperates with TFAP2C and BLIMP1 to upregulate germ cell and pluripotency genes, while repressing WNT signalling and somatic markers. Notably, PRDM14 targets are not conserved between mouse and human, emphasising the divergent molecular mechanisms of PGC specification. The effectiveness of degrons for acute protein depletion is widely applicable in various developmental contexts.

[1] Wellcome Trust/Cancer Research UK Gurdon Institute, Henry Wellcome Building of Cancer and Developmental Biology, Cambridge CB2 1QN, UK. [2] Physiology, Development and Neuroscience Department, University of Cambridge, Cambridge CB2 3EL, UK. [3] Wellcome Trust/Medical Research Council Cambridge Stem Cell Institute, University of Cambridge, Cambridge CB2 1QR, UK. [4] Institute for Systems Genetics, NYU Langone Health, New York, NY 10016, USA. ✉email: a.surani@gurdon.cam.ac.uk

Gametes develop from primordial germ cells (PGCs), the embryonic precursors, which are apparently specified in approximately week 2–3 human embryos[1,2]. While the requirement for BMP and WNT signalling for PGC specification is conserved between mouse and human[3–5], the gene regulatory network for hPGC fate has diverged[6]. Mouse PGCs (mPGCs) are specified by three core transcription factors (TFs): *Prdm1* (encoding BLIMP1), *Prdm14*, and *Tfap2c* (encoding AP2γ)[7,8], among which PRDM14 plays a central role; loss of *Prdm14* abrogates mPGC specification[9], while its overexpression is sufficient to induce mPGC fate in vitro[8].

During mPGC specification, PRDM14 induces upregulation of germline-specific genes, assists BLIMP1-mediated repression of somatic transcripts and initiates global epigenetic reprogramming[7,8,10,11]. PRDM14 also has a significant role in pre-implantation development[12], as well as pluripotency induction and maintenance in both mouse and human[13–16]. Indeed, *PRDM14* knockdown in hESCs led to a decrease in OCT4 levels and elevated expression of lineage markers[13,17,18].

Despite its critical function in mPGC specification, the role of PRDM14 in hPGC development remains uncertain, due to its low and potentially cytoplasmic expression in gonadal hPGCs[3]. Furthermore, a partial *PRDM14* knockdown suggested it might not be important for hPGC specification in vitro[19], within the TF network for hPGC specification that has diverged significantly from mouse[1,6,20]. In particular, SOX17 is a key determinant of hPGC fate, acting upstream of BLIMP1 and TFAP2C[3], but it is dispensable for mPGC development[21,22]. Understanding whether PRDM14 has a role in hPGC specification is critical towards gaining insights on the molecular divergence between mouse and human PGCs.

An inducible system for PRDM14 loss of function during hPGCLC specification from hESCs is critical, since PRDM14 is also vital for hESC pluripotency[13]. Accordingly, we combined auxin- or jasmonate-inducible degrons[23,24] with CRISPR/Cas9 genome editing[25] to achieve fast, comprehensive and reversible loss of endogenous PRDM14 protein. We reveal an indispensable role for PRDM14 in germ cell fate, since loss of function affects the efficiency of specification and results in an aberrant hPGCLC transcriptome. Notably, PRDM14 targets are not conserved between mouse and human, reflecting the evolutionary divergence in the molecular network for PGC specification. The study also illustrates the power of conditional degrons, which can be widely used to study TFs during cell fate determination.

## Results

**Detection of PRDM14 expression during hPGCLC specification.** To follow PRDM14 expression during hPGCLC specification, we appended Venus fluorescent protein to the C-terminus of endogenous PRDM14 (Fig. 1a) in the background of NANOS3-tdTomato hPGCLC-specific reporter[5]. PRDM14-T2A-Venus line served for flow cytometry and fluorescence-activated cell sorting (FACS) of PRDM14+ cells (Fig. 1b, c), while the fusion PRDM14-AID-Venus reporter was used to confirm subcellular localisation of PRDM14 (Fig. 1e), as well as for inducible protein degradation (see below). We detected Venus fluorescence in targeted hESCs and hPGCLCs but not in the parental control (Fig. 1b, c). Immunofluorescence (IF) confirmed co-localisation of Venus and PRDM14 in nuclei of both hESCs and hPGCLCs (Figs. 1e, 2a). Importantly, the majority of alkaline phosphatase (AP)+NANOS3-tdTomato+ hPGCLCs were PRDM14-Venus+ (Fig. 1c) and Venus+AP+ cells specifically expressed key germ cell markers (Fig. 1d).

We also assessed PRDM14 expression in human gonadal PGCs, and unlike some previous reports[3,5], we detected PRDM14 not only in the cytoplasm, but also in the nucleus of both male and female Wk7-9 hPGCs, (Supplementary Fig. 1A–D). Indeed, some gonadal sections showed only nuclear PRDM14 (Supplementary Fig. 1A, B). Nuclear localisation was also validated using IF on FACS-purified AP+cKIT+ gonadal hPGCs (Supplementary Fig. 1E). Based on the available data, PRDM14 localisation is unlikely to be a consequence of stage or sex of hPGCs. However, many more human foetal samples and stages will need to be tested in the future to determine the dynamics, if any, of PRDM14 localisation.

Next, we tracked PRDM14 expression dynamics upon hPGCLC induction by time-course IF, using SOX17 (Fig. 2 and Supplementary Fig. 2A), BLIMP1 (Supplementary Fig. 2B–D), TFAP2C (Supplementary Fig. 2F) and OCT4 (Supplementary Fig. 2A, B) to mark hPGCLCs. PRDM14 was uniformly expressed in 4i hESCs (Fig. 1e) that are competent for hPGCLC fate[3]. However, 12 h from the start of hPGCLC induction by cytokines, the levels of PRDM14 declined significantly in SOX17+ or BLIMP1+ cells (Fig. 2a), while the remaining PRDM14-Venus+ cells at 12 h were SOX2+, and thus likely represented neighbouring pluripotent cells (Supplementary Fig. 2E). The repression of PRDM14 in the putative hPGCLCs was however transient, since we observed specific re-expression of PRDM14 in ~25% of SOX17+ cells on day 1 (D1); the proportion of PRDM14+SOX17+ cells continued to increase progressively, reaching ~60% on D2, ~86% on D3, ~91% on D4 and ~93% on D5 (Fig. 2b). Many of the remaining SOX17+PRDM14− or BLIMP1+PRDM14− cells might belong to alternative lineages, as they lacked OCT4 expression (Supplementary Fig. 2A, B). Indeed, PRDM14 marked the majority of BLIMP1+OCT4+/SOX17+OCT4+ cells in D3–D5 embryoid bodies (Fig. 2c and Supplementary Fig. 2D).

Interestingly, OCT4, while specific to germ cells from D3 onwards, was detected in most cells of the EB at 12 h and on D1, as well as in many BLIMP1−/SOX17− cells on D2 (Supplementary Fig. 2A, B and ref. [3]). Of note, high SOX2 and PRDM14 levels persisted in the control cell aggregates in the absence of hPGCLC-inducing cytokines (Supplementary Fig. 2E).

We also assessed relative expression dynamics of PRDM14 and KLF4, a naive pluripotency-related TF known to be specifically expressed in hPGCs and hPGCLCs[3,26], but repressed in mouse germ cells[27]. Notably, KLF4 was not expressed in hPGCLC until D3, and KLF4 expression followed that of PRDM14 (Supplementary Fig. 2G).

Importantly, on D2, a higher proportion of BLIMP1+ than of SOX17+ cells expressed PRDM14 (Fig. 2b and Supplementary Fig. 2C), which agrees with the established order of SOX17 followed by BLIMP1 expression in hPGCLCs[3]. The majority of BLIMP1+ cells are SOX17+ and are therefore likely to progress to the upregulation of PRDM14. By contrast, some SOX17+ cells might be at an earlier stage lacking BLIMP1 expression and thus unlikely to have advanced far enough to express PRDM14.

The transient loss followed by re-expression of PRDM14 in cells undergoing hPGCLC specification suggests that PRDM14 might have a vital role in this cell fate decision. We pursued this possibility by using inducible degrons for an acute destabilisation of the PRDM14 protein.

**Inducible degrons allow efficient PRDM14 protein depletion.** Most inducible loss-of-function approaches act at the transcriptional or post-transcriptional levels and often suffer from slow and incomplete protein removal[28–31]. Since PRDM14 is one of the core pluripotency factors in hESCs[13], and exhibits dynamic changes during hPGCLC induction, conventional knockout approaches are unsuitable to study its functions in hPGCLCs. As PRDM14 expression commences within 24 h of hPGCLC induction, we decided to use conditional degrons[32] to achieve

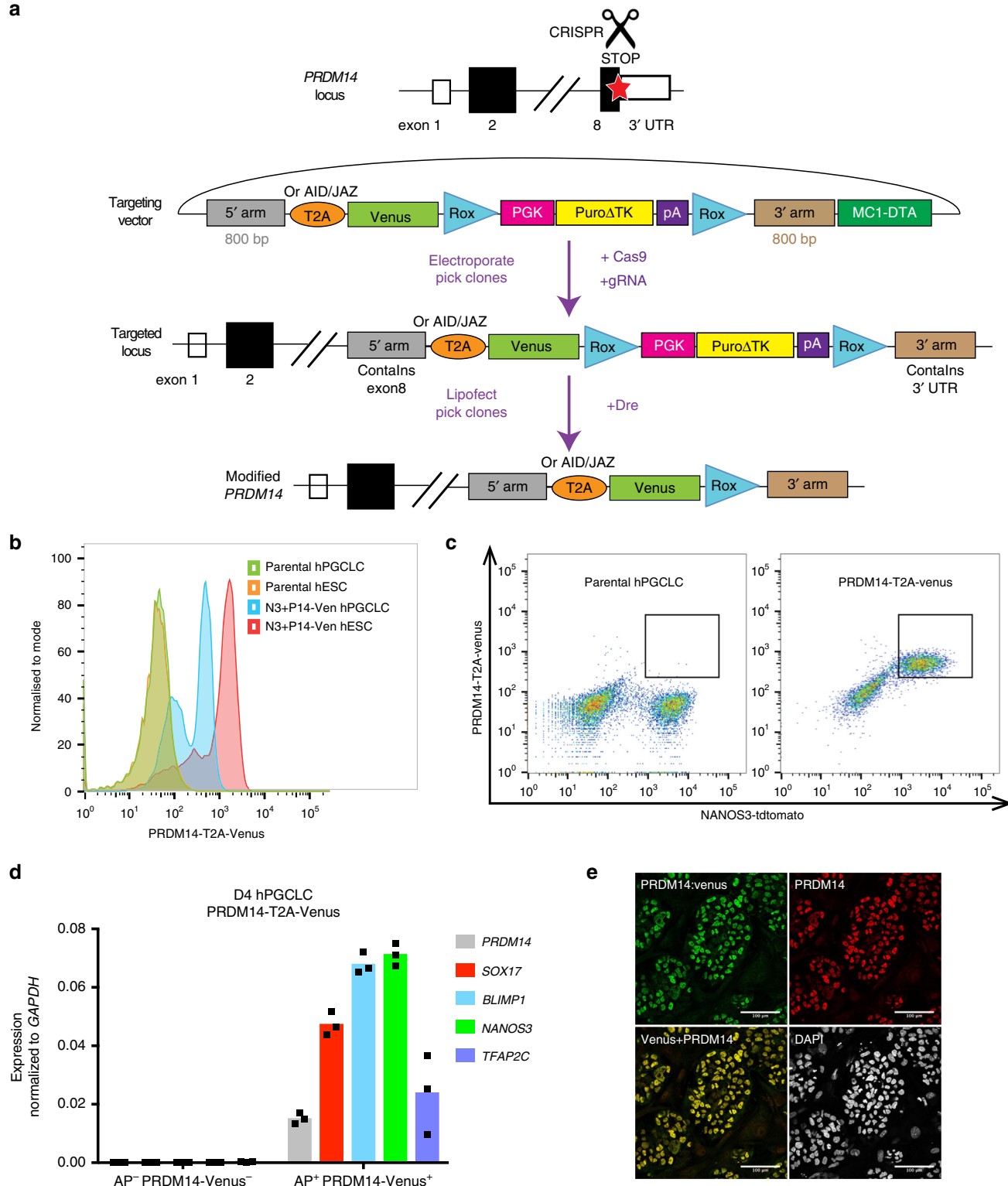

**Fig. 1 PRDM14-Venus knock-in reporters allow PRDM14 detection in hESCs and hPGCLCs. a** Scheme of CRISPR/Cas9-mediated *PRDM14* locus targeting to generate T2A-Venus, AID-Venus or JAZ-Venus reporter versions. 5' and 3' arms—homology sequences, T2A—self-cleaving peptide, AID—auxin-inducible degron, Venus—fluorescent gene, Rox—sequences for site-specific recombination recognised by the Dre enzyme, PGK-Puro—puromycin resistance gene under the control of PGK promoter, ΔTK—truncated thymidine kinase gene, MC1-DTA—diphtheria toxin fragment A gene under the control of MC1 promoter. Also see Fig. 3a. **b, c** Flow cytometry analysis showing Venus fluorescence in targeted hESCs and hPGCLCs compared with negative control. Note that Venus fluorescence predominantly coincides with NANOS3-tdTomato signal, which marks hPGCLCs. **d** qPCR analysis on sorted PRDM14-T2A-Venus⁺AP⁺ and double-negative cells from D4 EBs. Venus⁺AP⁺ population shows specific expression of germ cell markers. Data show results from three technical replicates (also see Source Data file). **e** IF analysis (representative of >10 experiments) of PRDM14-AID-Venus in competent hESCs showing co-localisation of PRDM14 and Venus fluorescence. Nuclei were counterstained by DAPI. Scale bar is 100 μm.

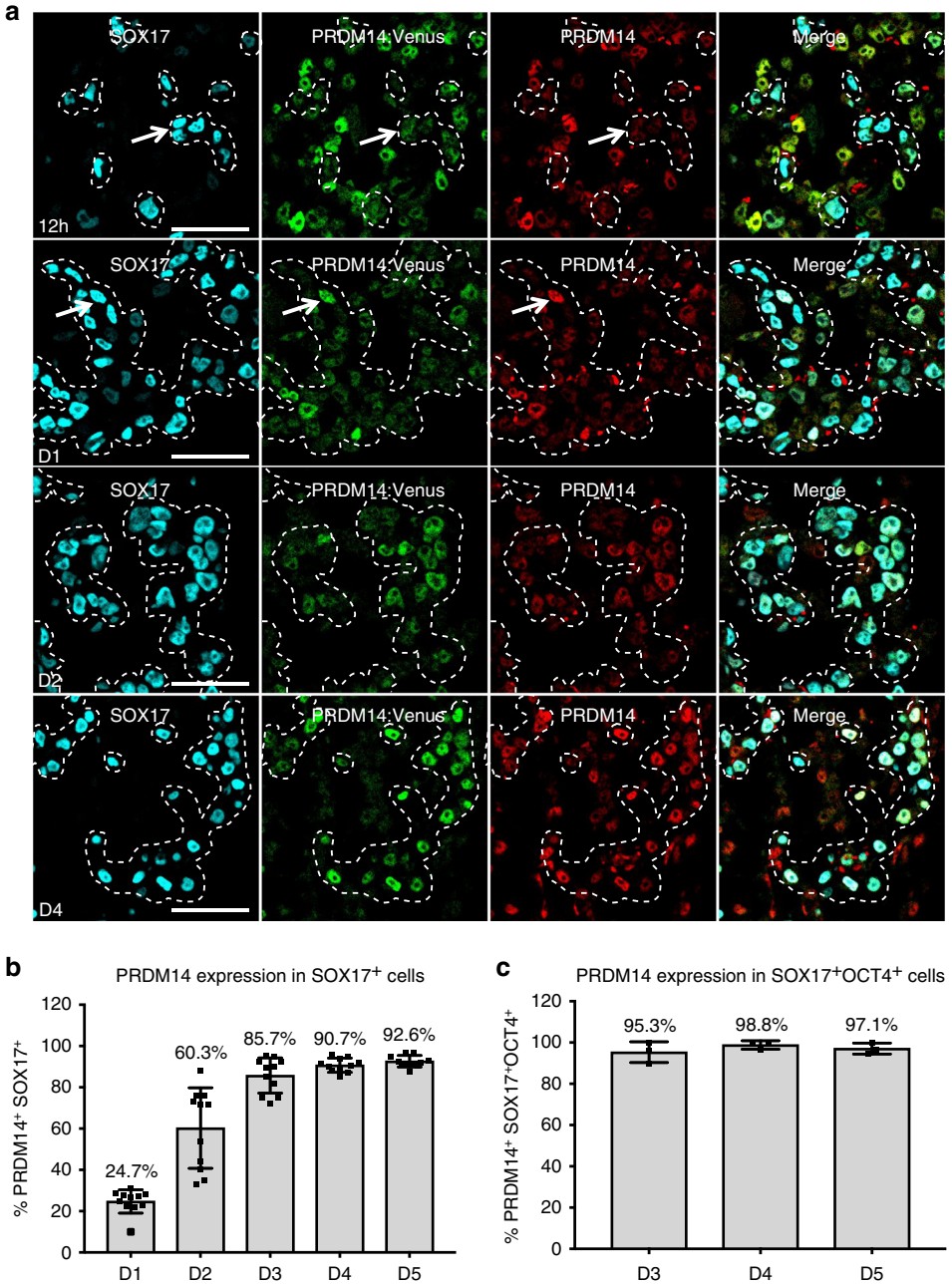

**Fig. 2 PRDM14-Venus knock-in reporter detects the dynamics of specific PRDM14 expression in hPGCLCs. a** Time-course IF analysis (representative of two hPGCLC inductions) showing PRDM14 expression in embryoid body (EB) sections throughout hPGCLC specification from PRDM14-AID-Venus fusion reporter cell line. hPGCLCs were induced by cytokines and EBs were collected at 12 h, and on D1-D5. Representative images for 12 h, D1, D2 and D4 are shown. hPGCLCs were marked by SOX17 and highlighted by a dashed line. Arrows show examples of SOX17+ cells that are PRDM14-negative at 12 h and PRDM14-positive on D1 of differentiation. Scale bar is 60 μm. **b** Quantification of results from **a** and Supplementary Fig. 2A, including the D1–D5 time points. Data show the percentage of PRDM14-Venus+SOX17+ cells as mean ± SD of $n = 11$ EB sections from two hPGCLC inductions. **c** Quantification of results from (Supplementary Fig. 2A) showing the percentage of PRDM14-Venus+SOX17+OCT4+ cells as mean ± SD of $n = 3$ EB sections from one hPGCLC induction.

fast, inducible and reversible PRDM14 depletion at the protein level.

Auxin and jasmonate-inducible degrons (AID and JAZ, respectively), are plant pathways for ligand-induced targeted protein degradation[23,24]. To harness AID and JAZ degrons in hESCs and hPGCLCs, we used CRISPR/Cas9 to tag the C-terminus of the endogenous PRDM14 with AID or JAZ degron sequences (see below) fused to Venus (Fig. 3a). Next, we employed the PiggyBac transposon system[33] to deliver codon-optimised transgenes encoding cognate hormone receptors from

rice (*Oryza sativa*): TIR1 for AID and COI1B for JAZ, allowing target degradation upon administration of auxin (indole-acetic acid, IAA) or coronatine (Cor), respectively[24,34]. Note that we fused COI1B with TIR1 F-box domain (resulting in Fb-TIR1-COI1B) to ensure better integration into human SCF (SKP1, CUL1 and F-box) E3 ubiquitin–ligase complex, as reported previously[24].

For AID, we chose the 44-amino-acid (residues 71–114) version of the *Arabidopsis thaliana* AID tag (*At*AID[44]) that was used successfully elsewhere[34,35]. Since the JAZ system has not

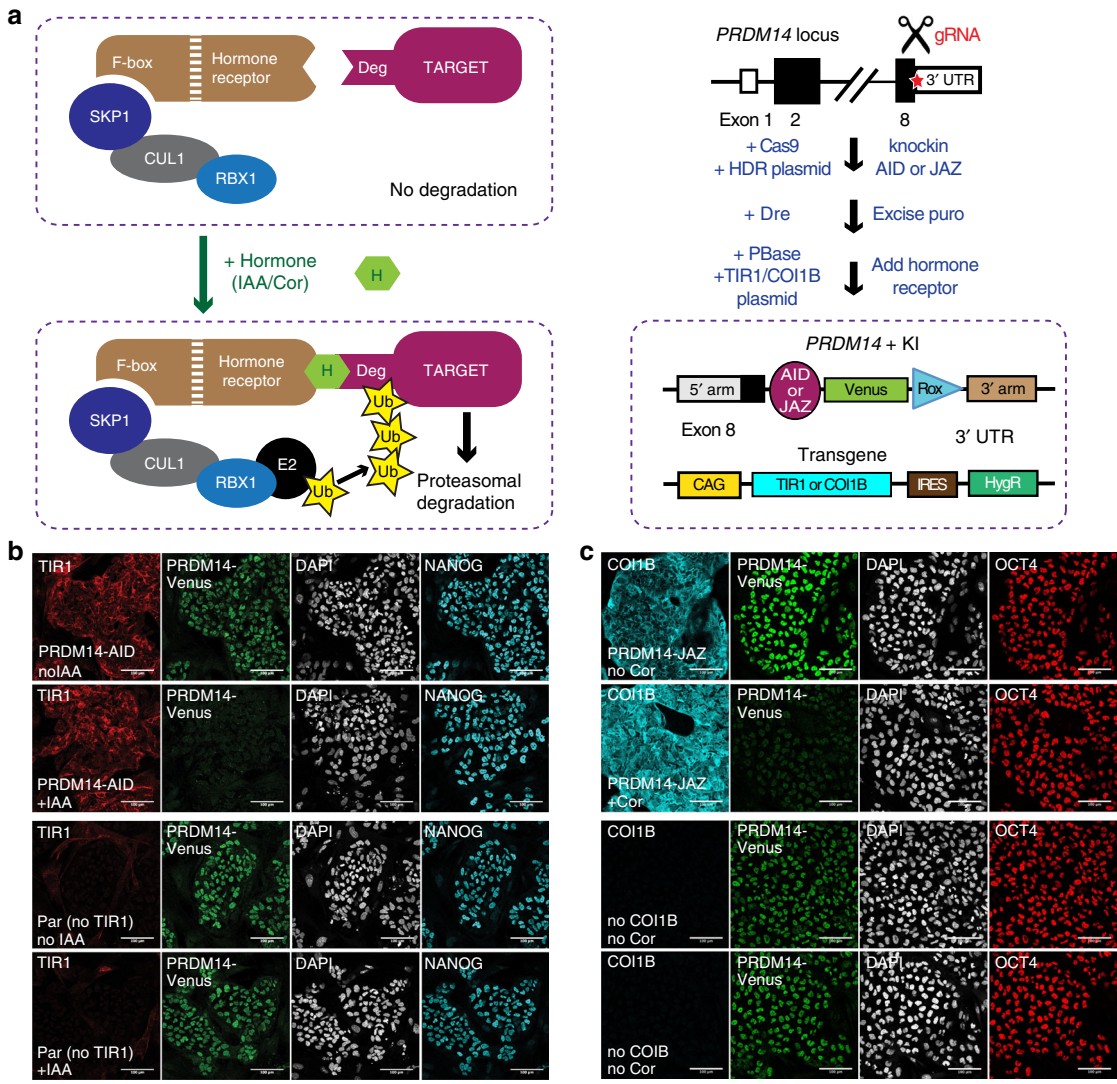

**Fig. 3 Auxin- and jasmonate-inducible degrons allow fast and homogenous PRDM14 protein depletion. a** Scheme of AID and JAZ systems and the derivation of PRDM14-AID-Venus and PRDM14-JAZ-Venus cell lines. SKP1, CUL1, RBX1 and F-box (TIR1 or COI1B) are the subunits of the SCF E3 ubiquitin–ligase complex. In the presence of IAA (for AID) or Cor (for JAZ) the target associates with the E3 ligase via a cognate hormone receptor (F-box protein); the recruited E2 ubiquitin-conjugating enzyme polyubiquitinates the target, which results in its proteasomal degradation[24,80]. For cell line generation, the cells were first transfected with two plasmids: (1) Cas9-encoding plasmid[25] with a gRNA sequence targeting the vicinity of the *PRDM14* stop codon; (2) HDR (homology-directed repair) template (also see Fig. 1a) containing the sequences to be appended by KI, flanked by two 800-bp homology arms; MC1-DTA was added to reduce random integration[5]. The knocked-in sequences were: AID (*At*AID[44]) or JAZ (*Os*JAZ[43]) followed by Venus and a positive/negative PGK-puromycin-ΔTK selection cassette flanked by Rox sites to allow excision by Dre[5]. Correct homozygous clones were transfected with PiggyBac constructs encoding the corresponding hormone receptor (TIR1 for AID and Fb-TIR1-COI1B for JAZ) under the control of the CAG promoter followed by an IRES and a hygromycin resistance gene. The homogeneity of myc-TIR1 or HA-COI1B expression in selected clones was checked by IF (see **b**, **c**). **b** IF (representative of three experiments) showing PRDM14, NANOG and TIR1 (myc-tagged) expression and PRDM14-AID-Venus depletion upon 25 min of IAA treatment. Nuclei were counterstained by DAPI. Scale bar is 100 μm. **c** IF (representative of three experiments) showing PRDM14-Venus, OCT4 and COI1B (HA-tagged) expression and PRDM14-JAZ-Venus depletion upon 2 days of Cor treatment. Nuclei were counterstained by DAPI. Scale bar is 100 μm.

been validated as extensively as AID, we performed additional optimisation experiments (Supplementary Fig. 3). We compared the best performing degron sequences from[24], namely *Os*JAZ33 (version 6-Os33) and NLS-*Os*JAZ33 (version 7), with a slightly longer *Os*JAZ43 degron sequence (without NLS). For this, HEK-293T cells were infected with viral constructs harbouring GFP fused to respective degron sequences and treated with Cor or DMSO (control). GFP depletion at 2, 4 and 24 h was monitored by flow cytometry (Supplementary Fig. 3A). This showed that the longer degron sequence (*Os*JAZ43) is superior to *Os*JAZ33 and

slightly better than NLS-*Os*JAZ33 in the efficiency and speed of GFP depletion (Supplementary Fig. 3A). This suggests that *Os*JAZ43 is more versatile and can allow depletion of both nuclear and cytoplasmic proteins.

To further test if JAZ could be efficiently used in our system, we generated hESCs expressing a PiggyBac-delivered construct (Supplementary Fig. 3B), where GFP was fused to the best *Os*JAZ43 degron sequence and an NLS, to better model the depletion of TFs such as PRDM14. The construct also harboured the Fb-TIR1-COI1B hormone receptor. GFP signal started

decreasing 3 h after Cor administration (Supplementary Fig. 3C) and 99% of cells were GFP-negative after one passage in Cor (Supplementary Fig. 3D). Based on these results, we used *OsJAZ*[43] and Fb-TIR1-COI1B combination to deplete PRDM14.

First, we tested AID- or JAZ-mediated PRDM14 depletion in hESCs by growing them with or without corresponding hormones and performing IF (Fig. 3b, c and Supplementary Fig. 4). PRDM14 and Venus fully colocalised and were both reduced upon the addition of IAA or Cor in hormone-sensitive, but not in parental lines (Fig. 3b, c and Supplementary Fig. 4A–D). Crucially, the AID system allowed reduction of Venus fluorescence to negligible levels, comparable with cells lacking the Venus knock-in (Supplementary Fig. 4C). Time-course IF in hESCs revealed a rapid onset of PRDM14-AID-Venus depletion within 10 min of IAA treatment, reaching homogeneity within 25 min (Fig. 3b and Supplementary Fig. 4E). In the case of the JAZ degron, however, residual PRDM14-JAZ-Venus signal was detectable even after 2 days of Cor treatment (Fig. 3c and Supplementary Fig. 4D). Both systems are nevertheless reversible[24,34], and IAA wash-off restored PRDM14 levels in less than 2 h (Supplementary Fig. 4F). Altogether, inducible degrons allow fast and reversible PRDM14 depletion, with AID being superior in terms of speed and efficiency.

**Inducible PRDM14 degradation reduces hPGCLC specification**. Next, we addressed the importance of PRDM14 for hPGCLC specification by adding IAA or Cor at the onset of hPGCLC differentiation (D0), followed by measuring the induction efficiency on D4 by recording the percentage of NANOS3-tdTomato[+]AP[+] cells (Fig. 4a). Notably, PRDM14 depletion using both approaches resulted in a significant reduction of hPGCLC induction efficiencies (by 70% and 30–60%, respectively) (Fig. 4b, c and Supplementary Fig. 5A–C). The effect was more pronounced in the case of the AID system, presumably due to its higher efficiency and faster kinetics of PRDM14 depletion. Therefore, we focused on the AID system to study PRDM14 further. Crucially, the observed phenotype was replicated using a PRDM14-AID in another hESC line (Supplementary Fig. 5D).

To further test AID efficacy in this context, we induced degradation of SOX17, a known hPGC and definitive endoderm (DE) regulator[3,20,36]. IAA addition fully abrogated hPGCLC and DE specification (Supplementary Fig. 5E–G), confirming the key roles of SOX17 in these lineages.

To establish the time when PRDM14 is essential for hPGCLC specification, we performed a time-course of IAA supplementation. We performed depletion of PRDM14 by adding IAA starting on D0, D1 or D2, followed by analysis on D4. We noted a strong phenotype with reduction in hPGCLC specification (Fig. 4d). By contrast, the addition of IAA on D3 had no detectable effect on the number of NANOS3-tdTomato[+]AP[+] cells (Fig. 4d). This suggests that either PRDM14 is dispensable after D2 or that its depletion at later time points does not affect hPGCLC numbers, although transcriptional or epigenetic consequences cannot be excluded. Altogether this points to an early critical role of PRDM14 in hPGCLC specification.

PRDM14 is highly expressed in human PGC-competent pluripotent cells, unlike in mice, where it is repressed at the equivalent stage[3,37]. Notably, however, PRDM14 depletion in 4i hESCs for one passage prior to hPGCLC induction (Supplementary Fig. 5H) had no effect on hPGCLC specification (Supplementary Fig. 5I), indicating no detectable involvement of PRDM14 in the maintenance of competence for hPGCLC fate. To determine if PRDM14 is required for the acquisition of competence, we induced competence for hPGCLCs in hESCs via pre-mesendoderm (preME), which transiently displays hPGCLC

potential[5]. However, when IAA was added to the preME medium and washed off before hPGCLC induction (Supplementary Fig. 5H), we observed no effect on hPGCLC specification efficiency (Supplementary Fig. 5J). Altogether, these data demonstrate that PRDM14 is important for hPGCLC specification but is dispensable for the acquisition and maintenance of the hPGCLC-competent state. Furthermore, this suggests a probable specific and direct requirement for PRDM14 in hPGCLC induction, which might not simply result from pluripotency disruption in the starting cell population.

**Restored PRDM14 levels rescue hPGCLC differentiation**. To confirm the specificity of the observed phenotype, we attempted to rescue the endogenous PRDM14 depletion with ectopic *PRDM14*, using the ProteoTuner system, whose kinetics is comparable with that of AID[38]. PRDM14 transgene was fused to a destabilisation domain (DD) and thus continuously degraded by the proteasome but stabilised by Shield-1 ligand[38,39]. Interestingly, overexpression of PRDM14 on D0, but not on D1 could fully rescue hPGCLC specification efficiency and even enhanced hPGCLC specification compared with "no IAA" control (Fig. 5a). This suggests that the transient PRDM14 repression observed at the onset of hPGCLC specification (Fig. 2b) is not prerequisite for germ cell induction and that despite low PRDM14 expression in SOX17/BLIMP1[+] cells on D0, it is apparently critical for hPGCLC induction.

We then asked if we could restore the endogenous PRDM14 levels by blocking auxin perception. To achieve this, we designed an AID-JAZ degron switch, where PRDM14-AID-Venus is degraded by IAA supplementation, while TIR1 is fused to JAZ and thus depleted by Cor (Fig. 5b). IF analysis in hESCs confirmed that simultaneous administration of both IAA and Cor desensitised hESCs to IAA via TIR1 degradation and replenished the PRDM14-Venus pool (Fig. 5c). In line with previous experiments, these cell lines also showed decreased hPGCLC specification efficiency upon IAA treatment (Fig. 5d). Crucially, in the presence of both IAA and Cor, hPGCLC differentiation efficiency was significantly restored (Fig. 5d). Interestingly, similar to PRDM14-DD overexpression above (Fig. 5a), significant rescue was only achieved on D0, but not on D1–D2 (Fig. 5e), confirming the early role of PRDM14 in hPGCLC specification. Overall, these data prove the causative role of PRDM14 depletion in hPGCLC induction phenotype and exemplify the use of orthogonal degron approaches for rescue design.

**PRDM14-deficient hPGCLCs show aberrant transcriptome**. Next, we examined the transcriptome of PRDM14-deficient hPGCLCs by performing RNA-sequencing (RNA-seq) on NANOS3-tdTomato[+]AP[+] hPGCLCs differentiated with or without IAA for 4 days from two hormone-sensitive clones and the parental control lacking TIR1. We also addressed the role of PRDM14 in pluripotency regulation, by analysing the transcriptome of AP[+] competent hESCs grown with or without IAA for one passage (3 days). Importantly, the parental ("no TIR1") control showed no differentially expressed genes (DEGs) in IAA-treated hESCs, and only two upregulated genes in hPGCLCs (*CYP1B1* and *LRAT*), indicating minimal non-specific effects of auxin on the target cells' transcriptome (Supplementary Data 1).

RNA-seq on PRDM14-depleted hESCs identified 106 upregulated and 64 downregulated genes (Fig. 6a and Supplementary Data 1). Crucially, our findings concur with a previous study on PRDM14 in different hESC lines and culture conditions[13] (Fig. 6c), with downregulation of pluripotency genes (e.g. *OCT4*, *NANOG*, *TDGF1* and *GAL*), and upregulation of differentiation

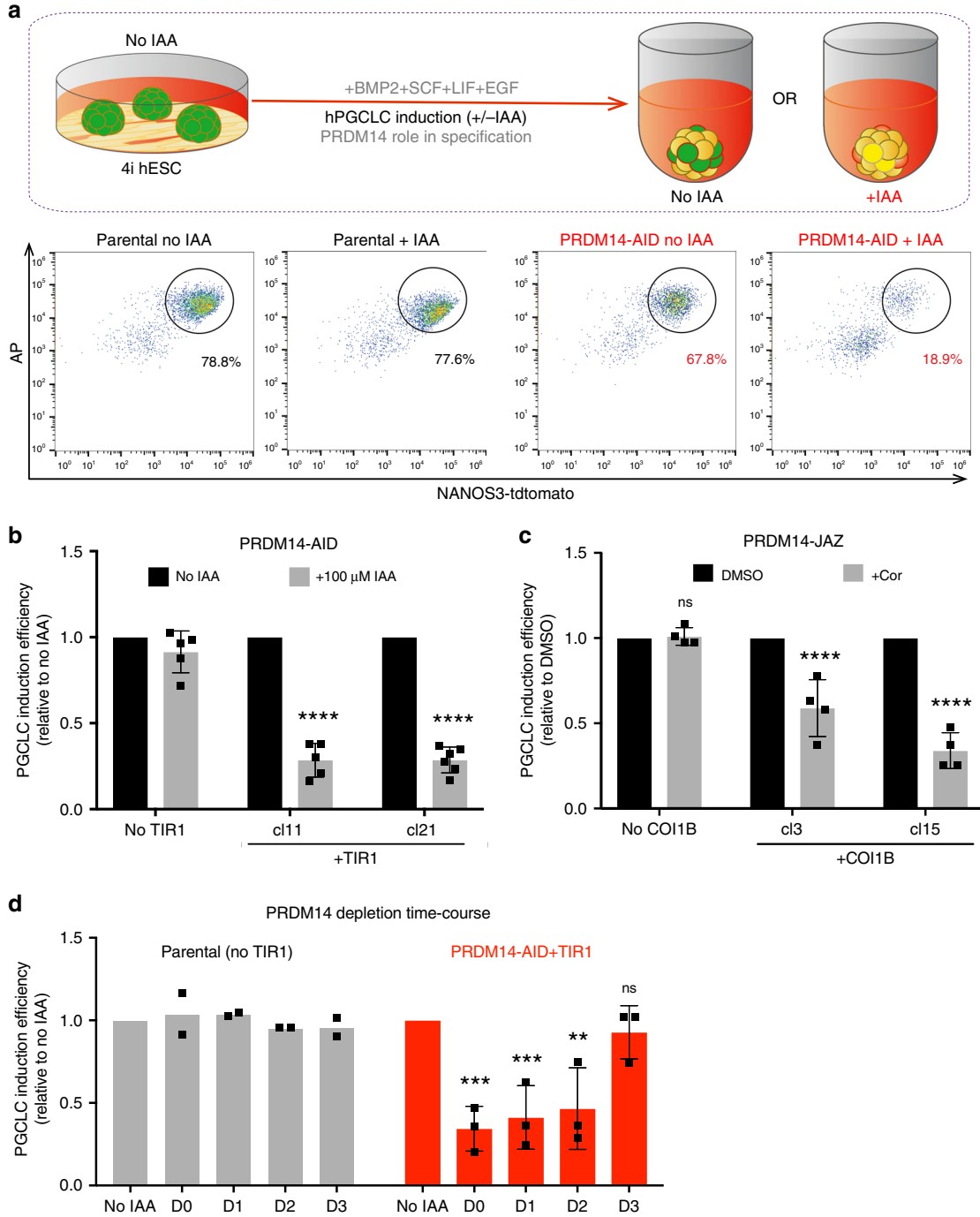

**Fig. 4 PRDM14 depletion using AID or JAZ degrons reduces hPGCLC specification efficiency. a** Experimental design and representative flow cytometry plots showing reduced hPGCLC induction in hormone-sensitive clones, but not in the parental control. hPGCLCs were defined as NANOS3-tdTomato[+]AP[+] cells. **b** Quantification of flow cytometry results for PRDM14-AID-Venus hPGCLCs induced with or without IAA (100 μM). The induction efficiency of untreated cells ("no IAA") was set to 1. Data show mean ± SD for $n = 5$ for cl11 and "no TIR1" control and $n = 6$ for cl21, ns not significant ($P = 0.1308$), ****$P < 0.0001$ (Two-way ANOVA followed by Sidak's multiple comparison test). **c** Quantification of flow cytometry results for PRDM14-JAZ-Venus hPGCLCs. The induction efficiency of untreated cells ("no Cor") was set to 1. Data show mean ± SD for $n = 4$ independent experiments per clone, ns not significant ($P = 0.9979$), ****$P < 0.0001$ (Two-way ANOVA followed by Sidak's multiple comparison test). **d** Time-course of PRDM14 depletion. IAA (500 μM) was added on indicated days of hPGCLC induction from PRDM14-AID-Venus clones. Note that IAA was not washed off until the end of differentiation (D4), when hPGCLC induction efficiency was assessed by flow cytometry. Data show mean ± SD of $n = 3$ for PRDM14-AID or mean of $n = 2$ for "no TIR1" control. Statistical significance was analysed by two-way ANOVA followed by Sidak's multiple comparison test. For PRDM14-AID: no IAA vs. D0 ***($P = 0.0003$), no IAA vs. D1 ***($P = 0.0008$), no IAA vs. D2 **($P = 0.0018$), no IAA vs. D3 ns (not significant, $P = 0.9614$); for "no TIR1": all ns not significant ($P > 0.99$).

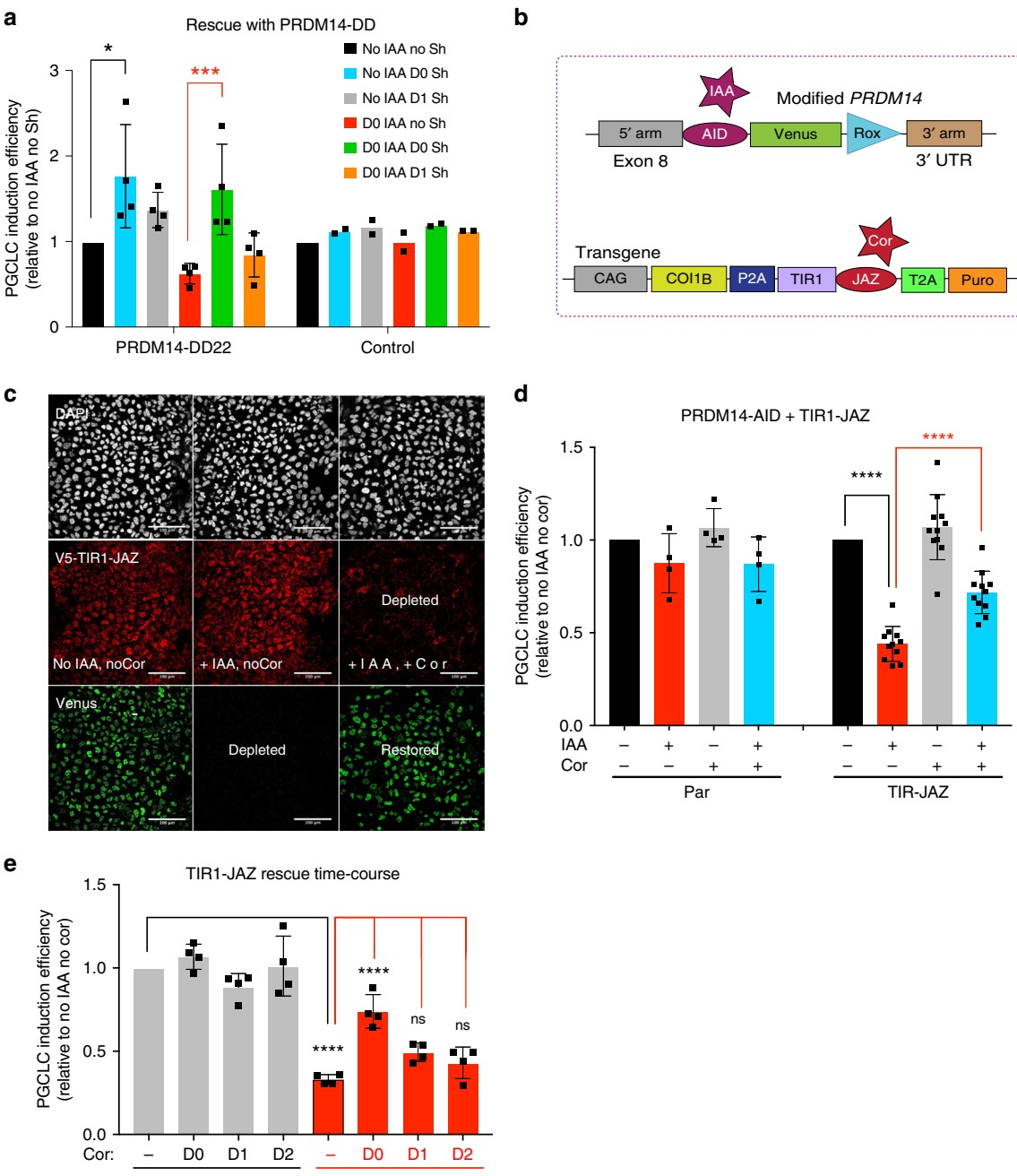

**Fig. 5 Ectopic PRDM14 expression or desensitisation to auxin rescue hPGCLC specification. a** IAA and/or Shield-1 (Sh) were added on indicated days of differentiation to deplete the endogenous PRDM14 or stabilise the PRDM14-DD transgene, respectively. "Control" cell line lacks both the AID knock-in and the PRDM14-DD transgene. Data show mean ± SD for $n = 4$ (PRDM14-DD22) or mean of $n = 2$ (control) independent experiments. Statistical significance was analysed by two-way ANOVA followed by Sidak's multiple comparison test; for PRDM14-DD22: D0 IAA no Sh vs. no IAA no Sh not significant, ($P = 0.4159$), D0 IAA no Sh vs. no IAA D0 Sh ***($P = 0.0001$), D0 IAA no Sh vs. no IAA D1 Sh *($P = 0.0125$), D0 IAA no Sh vs. D0 IAA D0 Sh ***($P = 0.0008$), D0 IAA no Sh vs. D0 IAA D1 Sh not significant ($P = 0.8662$); for control: all not significant ($P > 0.97$). **b** Scheme of the double AID/JAZ degron design. Endogenous PRDM14 is under the control of the AID degron as above, while the stability of TIR1 hormone receptor is controllable by the JAZ degron. **c** AID/JAZ validation by IF in hESCs (representative of three experiments). Simultaneous IAA and Cor supplementation depletes TIR1 and restores PRDM14, as shown by IF staining for PRDM14-AID-Venus and TIR1(V5-tagged). Nuclei were counterstained by DAPI. Scale bar is 100 μm. **d** Desensitisation to auxin via the AID/JAZ degron switch rescues hPGCLC induction efficiency. Data show mean ± SD for $n = 4$ (no TIR1 control) or $n = 11$ (TIR-JAZ) biological replicates, ****$P < 0.0001$ (Two-way ANOVA followed by Sidak's multiple comparison test). **e** Time-course of desensitisation to auxin via the AID/JAZ degron switch. Note that hPGCLC induction efficiency is only significantly rescued if Cor is supplemented from D0. Addition of Cor alone (without IAA) does not alter hPGCLC specification efficiency. Data show mean ± SD for $n = 4$ biological replicates, ns not significant (D0 IAA no Cor vs. D0 IAA D1 Cor $P = 0.3982$; D0 IAA no Cor vs. D0 IAA D2 Cor $P = 0.9832$), ****$P < 0.0001$ (Two-way ANOVA followed by Sidak's multiple comparison test).

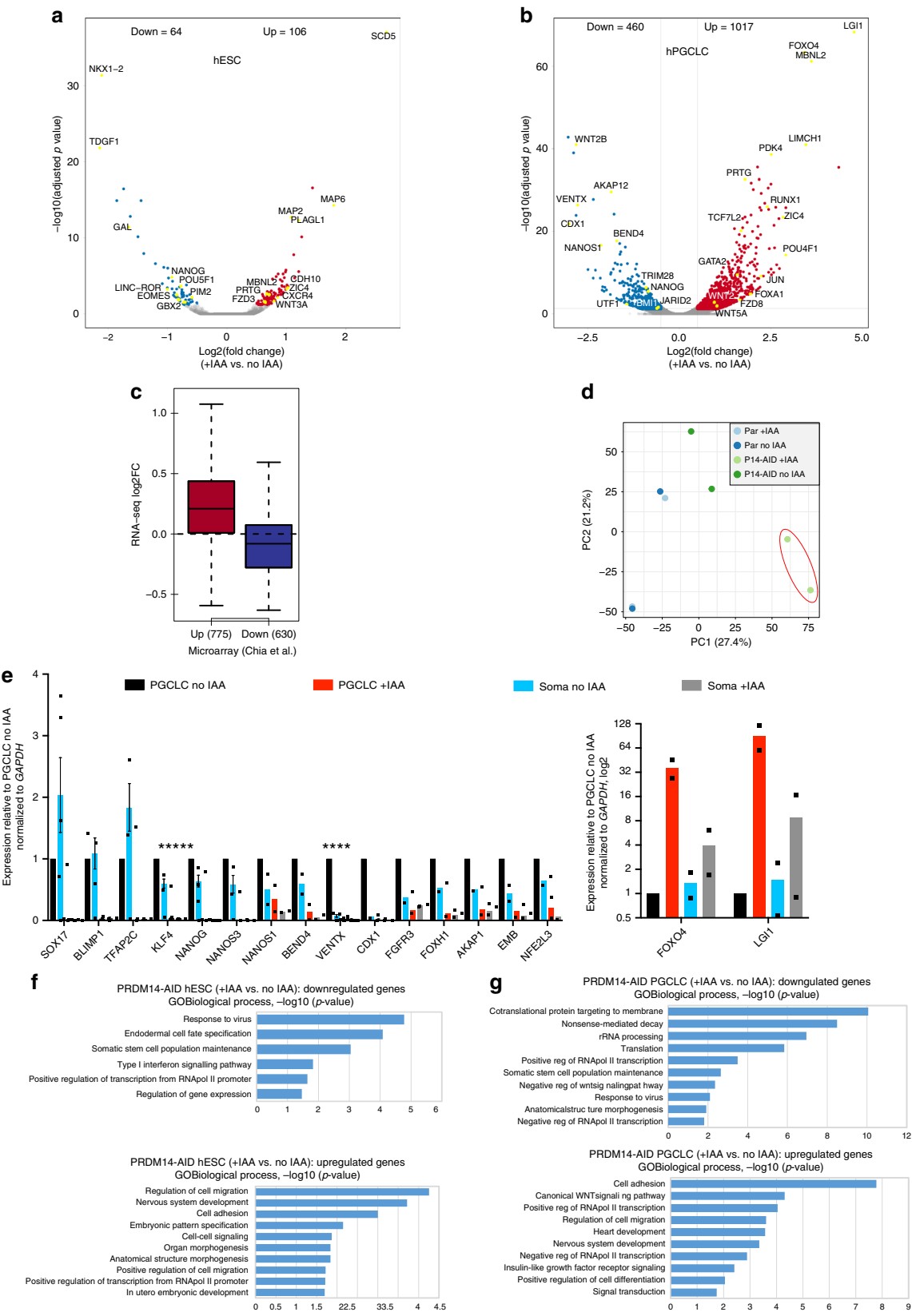

markers, particularly of the neural lineage (e.g. *MAP2*, *MAP6* and *SOX2*) (Fig. 6a, Supplementary Fig. 6E and Supplementary Data 1). Gene ontology (GO) analysis reflected reduction in self-renewal (enriched downregulated term "somatic stem cell maintenance") and upregulation of pro-differentiation genes

(enriched terms "nervous system development" and "embryonic pattern specification") (Fig. 6f).

Notably, there was a stronger response to PRDM14 loss in hPGCLCs, with 1017 upregulated and 460 downregulated genes in "+IAA" vs "no IAA" samples (Fig. 6b), and only a small

**Fig. 6 RNA-seq reveals transcriptional changes in PRDM14-deficient hESCs and hPGCLCs. a** Volcano plot showing down- (log2FC < (−0.5), padj < 0.05) and upregulated (log2FC > 0.5, padj < 0.05) genes in hPGC-competent hESCs upon PRDM14 loss. PRDM14-AID hESCs were treated with IAA for one passage (3 days) and sorted as AP⁺ cells for RNA-seq along with "no IAA" control. **b** Volcano plot showing down- (log2FC < (−0.5), padj < 0.05) and upregulated (log2FC > 0.5, padj < 0.05) genes in hPGCLCs upon PRDM14 loss. PRDM14-AID hPGCLCs were specified with or without IAA for 4 days and sorted as NANOS3-tdTomato⁺AP⁺ cells for RNA-seq along with "no IAA" control. **c** Boxplot overlaying down- (log2FC < (−0.5), padj < 0.05) and upregulated (log2FC > 0.5, padj < 0.05) genes upon *PRDM14* knockdown in conventional hESCs (microarray data from[13]) and RNA-seq analysis in PRDM14-AID-depleted PGC-competent (4i) hESCs from this study. The distance from the bottom to the upper line of the box spans the interquartile range (IQR), the horizontal line in the box indicates the median of the dataset, while the whiskers show ±1.5× IQR. Note that the majority of differentially expressed genes show consistent changes. **d** Principal component analysis (PCA) of hPGCLC RNA-seq samples. PRDM14-depleted hPGCLCs (cl11 and cl21; highlighted in red) separate from the other samples along principal component 1 (PC1). **e** qPCR validation of selected DEGs from RNA-seq. hPGCLCs from cl11 and cl21 were sorted as NANOS3-tdTomato⁺AP⁺ cells, while soma denotes the double-negative population from the same experiments. Strongly upregulated genes are plotted separately. Data show gene expression relative to no IAA hPGCLCs and normalised to *GAPDH*, mean ± SEM of *n* = 5 for *SOX17* and *VENTX*, mean ± SEM of *n* = 3 independent experiments for *NANOS3*, *BLIMP1*, *TFAP2C*, *KLF4* or *n* = 2 for other genes. Statistical significance of hPGCLC no IAA vs. +IAA was assessed by multiple unpaired *t*-tests yielding two-tailed *P* values: KLF4 **(*P* = 0.0054), NANOG **(*P* = 0.0053), NANOS3 *(*P* = 0.0452), VENTX ****(*P* < 0.0001). The parental ("no TIR1") control is shown in Supplementary Fig. 6D. **f** Gene ontology (GO) analysis on down- and upregulated genes in PRDM14-deficient hESCs. Top non-redundant GO terms are shown as −log10(*p* value). **g** GO analysis on down- and upregulated genes in PRDM14-deficient hPGCLCs. Top non-redundant GO terms are shown as −log10(*p* value). For complete list of GO terms see Supplementary Data 6.

overlap between the affected genes in hESCs and hPGCLCs (Supplementary Data 1). Principal component analysis (PCA) and unsupervised hierarchical clustering of hPGCLC transcriptome confirmed that parental control, irrespective of IAA treatment, clustered with untreated hormone-sensitive clones, and away from the PRDM14-depleted hPGCLCs that clustered together (Fig. 6d and Supplementary Fig. 6A). Crucially, PRDM14-deficient hPGCLCs showed downregulation of many key hPGC- and pluripotency-related genes, including *UTF1*, *NANOG*, *NANOS1*, *LIN28A* and *TRIM28*. GO analysis showed enrichment of protein biosynthesis-related processes, somatic stem cell population maintenance and negative regulation of WNT signalling (Fig. 6h). Gene set enrichment analysis (GSEA) against 123 core PGC genes defined previously[5], showed lower enrichment score of mutant hPGCLCs compared with the control (Supplementary Fig. 6B). The levels of key hPGC regulators, *SOX17* and *PRDM1*, however did not change (Fig. 6e and Supplementary Fig. 6D), which is consistent with their upregulation prior to *PRDM14* in nascent hPGCLCs (Fig. 2, Supplementary Fig. 2A, B and ref. [3]). Crucially, top changes in gene expression were also confirmed by qPCR using independent hPGCLC inductions (Fig. 6e and Supplementary Fig. 6D). qPCR analysis also showed downregulation of *KLF4* in PRDM14-depleted hPGCLCs (Fig. 6e), which, together with the observation that PRDM14 expression precedes that of KLF4 (Supplementary Fig. 2G), might suggest that PRDM14 could be upstream of KLF4 in hPGCLCs. Ectopic *PRDM14* could partially revert the transcriptional changes in mutant hPGCLCs (Supplementary Fig. 6C), in line with rescued hPGCLC specification phenotype (Fig. 5a).

GO analysis on genes derepressed in PRDM14-deficient hPGCLCs identified enrichment of terms related to WNT signalling, as well as heart and nervous system development (Fig. 6g). A similar response occurs in hPGCLCs specified without *BLIMP1* or *TFAP2C*[20,26]. Indeed, pairwise comparisons revealed a significant overlap between both up- and downregulated targets, suggesting potential combinatorial roles of PRDM14 with BLIMP1 and TFAP2C. Accordingly, 115 genes were downregulated in all three mutants (including germ cell markers *UTF1*, *NANOG*, *AKAP12*, *NANOS1* and *TRIM28*), while 281 shared upregulated genes were mostly implicated in morphogenesis, WNT signalling, as well as cell migration and adhesion (Supplementary Fig. 6F–G and Supplementary Data 2).

**PRDM14 regulates gene expression through promoter binding.** To identify direct targets of PRDM14 in hPGCLCs and hESCs, we performed chromatin immunoprecipitation (ChIP) with high-throughput sequencing (ChIP-seq). We took advantage of the PRDM14-AID-Venus line to use a ChIP-grade anti-GFP antibody for PRDM14-Venus immunoprecipitation, which proved more efficient than anti-PRDM14 antibodies (Supplementary Fig. 7A). Importantly, auxin treatment led to a significant loss of signal (Supplementary Fig. 7A), which allowed us to use ChIP-seq on IAA-treated hESCs as a control.

Overall, ChIP-seq identified 6486 consensus PRDM14 peaks: 4206 in hPGCLCs and 3319 in hESCs (Supplementary Data 3). *k*-means clustering partitioned the dataset into five distinct clusters (Fig. 7a), separating peaks specific to hESCs (742 peaks, cluster 4) or hPGCLCs (1331 peaks, cluster 5) or shared between the two (clusters 1–3). Note that the strongest ChIP-seq enrichment persisted in IAA-treated hESCs (clusters 1–2) with 1728/3319 hESC binding sites detectable in hESC + IAA, albeit with significantly lower signal (Fig. 7a and Supplementary Data 3). Such persistent regions might be more tightly bound, and/or require longer auxin exposure to eliminate PRDM14 binding completely.

Notably, PRDM14 in both hESCs and hPGCLCs predominantly binds within 1 kb of transcription start sites (TSS) (Supplementary Data 3), with ~30% of peaks spanning annotated promoters (Fig. 7b), which agrees with a previous ChIP-seq in hESCs[13]. By contrast, PRDM14 binds predominantly to distal genomic regions in mice, and only 4–10% of peaks are located within 1 kb from the TSS in mESCs[16,40].

Motif analysis[41] confirmed that the conserved PRDM14 motif was top ranking in both hESCs and hPGCLCs (Fig. 7c). Notably, TFAP2C motif was the second most enriched within hPGCLC-specific targets, suggesting cooperation between the two factors (Fig. 7c), consistent with the overlapping targets from RNA-seq (Supplementary Fig. 6F). We also found significant enrichment of BLIMP1 and SOX motifs in hPGCLCs, whereas unique hESC peaks were enriched in the OCT4-SOX2 motif (Fig. 7c). Altogether, this indicates that PRDM14 co-occupies genomic targets with core TFs specific to hPGCs or pluripotent cells, respectively.

Next, we correlated ChIP-seq peaks mapping within 100 kb of the TSS with the corresponding changes in gene expression from RNA-seq (Fig. 7d, Supplementary Fig. 6H and Supplementary Data 3). In hPGCLCs, 314 peaks were associated with 168 (17%) upregulated genes and 93 peaks with 47 (10%) downregulated genes (Supplementary Data 3). GO analysis confirmed that genes related to WNT signalling, heart and brain development are among the direct PRDM14 targets (Supplementary Fig. 7B). The

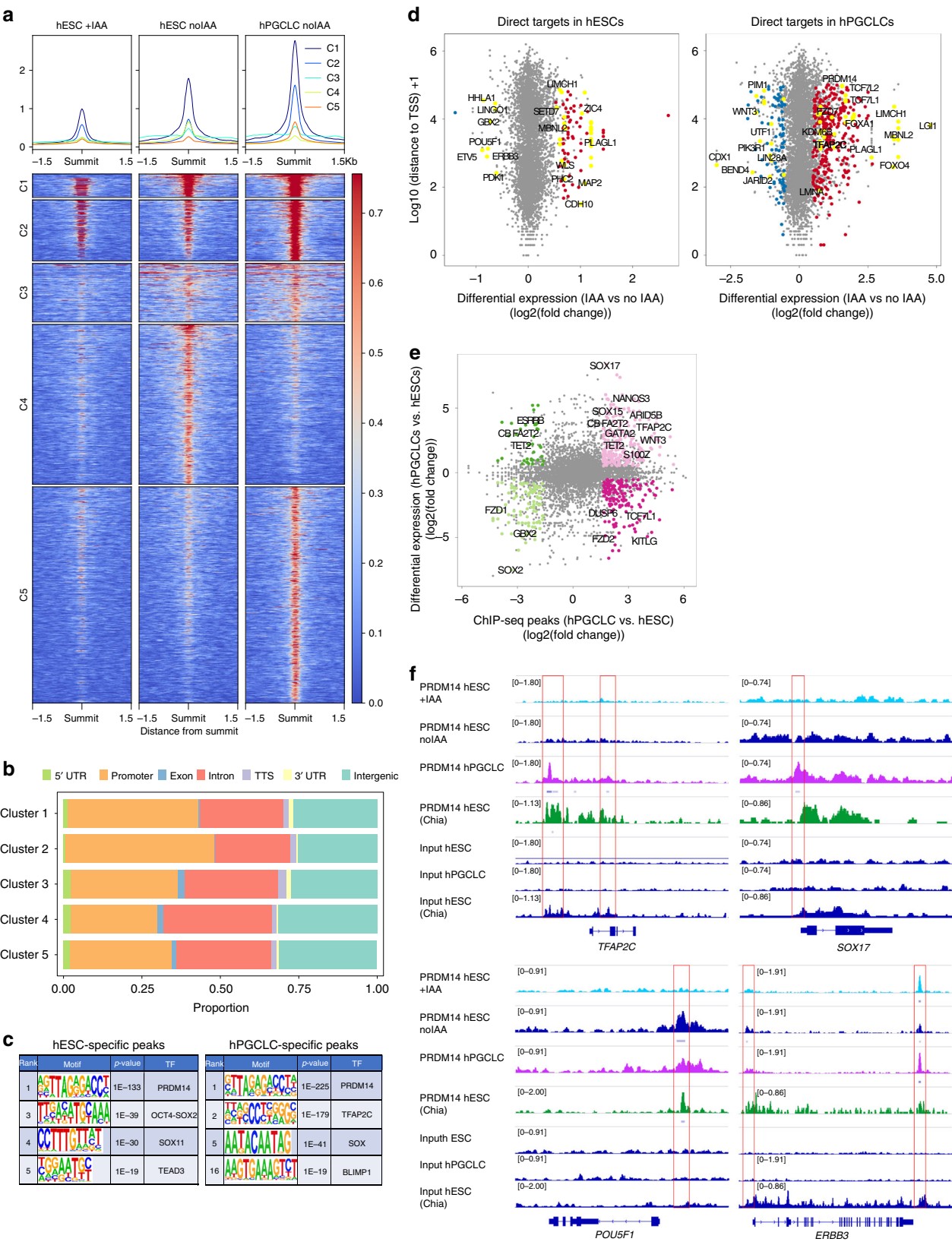

hESC-specific targets included *SOX2, ERBB3* and *GBX2*, while hPGCLC-specific peaks were found near *SOX17, TFAP2C* and *NANOS3* among others (Fig. 7e, f). Whereas PRDM14 binds to the regulatory elements of *SOX17*, PRDM14 depletion did not significantly affect SOX17 expression in hPGCLCs (Fig. 6e and

Supplementary Data 1) presumably because other factors sustain its expression.

**Distinct molecular functions of PRDM14 in mouse and human.** Next, we asked if the molecular roles of PRDM14 are

**Fig. 7 ChIP-seq analysis shows hESC- and hPGCLC-specific PRDM14 binding patterns. a** Consensus PRDM14-bound peaks (6486) are separated into five clusters by *k*-means clustering. The ChIP signals (counts per million per 10-bp bin) of each peak are shown in the clustered heatmaps (bottom panel). The profile plots (top panel) show the average ChIP signals of each cluster. **b** Distribution of clustered PRDM14 peaks in the genome. Note that PRDM14 predominantly binds to promoters. **c** Top-ranking motifs identified by HOMER de novo motif analysis within hESC- and hPGCLC-specific PRDM14 peaks (FC > 3, *p* value < 0.0001). **d** Scatter plots highlighting examples of direct PRDM14 targets in hESCs and hPGCLCs. Direct targets were defined as genes with at least one PRDM14 ChIP-seq peak (*q* value < 0.05) within 100 kb of the TSS and showing significant changes in expression upon IAA treatment (log2FC < (−0.5) or >0.5 and padj < 0.05). **e** PRDM14 binds to regulatory regions of key hPGC-related genes. The scatter plot shows differential PRDM14 peaks (FC > 3, *p* value < 0.0001) correlated to genes differentially expressed between hESCs and hPGCLCs (log2FC < (−0.5) or >0.5 and padj < 0.05). **f** Examples of loci bound by PRDM14 in hPGCLCs and hESCs. Note that the peak downstream of *ERBB3* is preserved in hESCs after IAA treatment. PRDM14 peaks in indicated samples were visualised using IGV software. PRDM14 ChIP-seq dataset from ref. [13] was added for comparison.

conserved between mouse and human, by comparing the protein-coding transcriptomes of PRDM14-AID hESCs and hPGCLC, with the equivalent mouse *Prdm14*$^{−/−}$ cells[11] (Supplementary Fig. 8A, B and Supplementary Data 4). PGC-competent mouse epiblast-like cells showed very few DEGs since *Prdm14* is repressed in these cells[11]. Only two genes (*CDH4* and *HS6ST2*) were derepressed in both mESCs and hESCs, and three genes (*DNMT3B*, *SPRY4* and *FHL1*) showed opposite changes, probably reflecting the broader differences between mESCs and hESCs[42]. Comparison of PRDM14-depleted hPGCLCs with D6 *Prdm14*$^{−/−}$ mPGCLC identified a limited subset of potentially conserved PRDM14 targets: 36 up- and 13 downregulated genes (Supplementary Fig. 8A), but none have hitherto been implicated in germ cell biology. Furthermore, the expression of 39 DEGs was anti-correlated in hPGCLC and mPGCLCs (Supplementary Fig. 8A).

In mPGCs, PRDM14 is key to epigenetic reprogramming, ensuring DNA demethylation through repression of *Uhrf1* and loss of H3K9me2 through *Ehmt1* downregulation[9]. hPGCs display similar hallmarks of epigenetic reprogramming, associated with repression of *UHRF1* and *EHMT2* (ref. [26]). While by RNA-seq we did not detect changes in *UHRF1* or *EHMT2* expression in PRDM14-deficient hPGCLCs, we verified UHRF1 and H3K9me2 levels by IF (Supplementary Fig. 8C, D), to exclude regulation at the protein level, as observed in other contexts[43–45]. This showed an increase in the proportion of UHRF1$^+$ proliferating D4 hPGCLCs upon PRDM14 depletion, potentially suggesting a conserved regulation (Supplementary Fig. 8C, G). The effect was, however, heterogeneous and there was no significant difference in mean fluorescence intensities of UHRF1 in proliferating hPGCLCs induced with or without IAA (Supplementary Fig. 8E). We did not detect significant retention of H3K9me2 in PRDM14-deficient hPGCLCs (Supplementary Fig. 8D, F, H). Overall this could suggest that PRDM14 is less critical for epigenetic reprogramming in hPGCs than it is in the mouse. However, D4 hPGCLCs model pre-migratory germ cells[3], which might be too early to detect potential defects in epigenetic resetting caused by PRDM14 depletion.

PRDM14 alone is sufficient to induce mPGCLCs fate[8]; however, this was not the case for hPGCLC specification using either doxycycline-inducible or ProteoTuner systems (Supplementary Fig. 8I), consistent with the observation that upregulation of SOX17, BLIMP1 and TFAP2C precede PRDM14 expression (Fig. 2 and Supplementary Fig. 2). Altogether PRDM14 evidently plays an important role shortly after the initiation of hPGCLC fate, but its function is distinct from that in mPGCs.

## Discussion

Using two acute protein depletion strategies, combined with rescue, transcriptomic and ChIP-seq experiments, we demonstrate that PRDM14 is required for hPGCLC specification and represses somatic differentiation while promoting germline fate (Supplementary Fig. 9). Strikingly, the molecular function of

PRDM14 in the human germline has diverged significantly compared with mPGCs. Indeed, the sets of targets regulated by PRDM14 in the two species are vastly different (Supplementary Fig. 8A, B) and, unlike in the mouse[8], PRDM14 alone cannot induce hPGCLCs (Supplementary Fig. 8D). The regulatory network for hPGC specification is altogether distinct from that in mice, with the recently established critical role of SOX17, and the notable repression of SOX2 (refs. [3,20]). Our study of PRDM14 provides further insights on the divergence of the molecular basis of germ cell fate determination in mouse and human.

PRDM14 is specifically upregulated in the nucleus of nascent hPGCLC, its expression follows that of SOX17, BLIMP1 and TFAP2C, but precedes the upregulation of KLF4 (Fig. 2 and Supplementary Fig. 2). PRDM14 depletion strongly reduces the efficiency of germ cell specification (Fig. 4) and results in an aberrant transcriptome of the PRDM14-deficient hPGCLCs (Fig. 6). Conversely, PRDM14 overexpression on D0 of induction rescues and even enhances hPGCLC specification (Fig. 5). Since hPGCLC induction is probably not a fully synchronous process, it is possible that PRDM14 overexpression in cells that start upregulating SOX17 and BLIMP1 might enhance their propensity to adopt hPGCLC fate through accelerating the expression of PRDM14 target genes, which perhaps helps consolidating germ cell identity.

The effects of PRDM14 depletion during hPGCLC specification, resemble those observed upon the loss of *TFAP2C* or *BLIMP1* (Supplementary Fig. 6F). Furthermore, ChIP-seq analysis revealed high enrichment of TFAP2C and BLIMP1 motifs within hPGCLC-specific PRDM14 targets (Fig. 7c), which suggests combinatorial roles of the three regulators during the induction of hPGC-specific genes and the repression of somatic markers. However, unlike in mPGCs[11,46], many prominent germ cell genes, such as *TFAP2C, PRDM1* and *DND1*, are not the targets of PRDM14 in hPGCLCs. Furthermore, while the PRDM14 motif is conserved, PRDM14 binds predominantly to gene promoters in hESCs and hPGCLCs[13] (Fig. 7b), but to distal regulatory elements in mice[16,40]. The notable lack of overlap might be dictated by different protein partners or by the divergence of the protein itself, although there is significant functional conservation of the protein, as human PRDM14 rescues the lack of its mouse orthologue in mESCs[15].

The context-dependent roles of PRDM14 are also exemplified by distinct set of target genes in hESCs, where it cooperates with OCT4 and SOX2 to sustain pluripotency, and limits neuronal differentiation (Fig. 6a, Supplementary Figs. 6E, 7C and 9). This and other studies highlighted the divergent roles of PRDM14 in mouse and human pluripotent cells[13,15,16,40]. Furthermore, there are significant human–mouse differences during early embryo development, at the time of PGC specification[47,48]; altogether these differences likely contribute to the species-specific modes of PGC specification. Notably, we found an even more pronounced transcriptional phenotype upon PRDM14 loss in hPGCLCs than in hESCs (Fig. 6). Note that hESCs were, however, in a 'steady state' of self-renewal, while hPGCLCs were undergoing cell fate

transitions at the time of PRDM14 depletion. The persistence of PRDM14 at some loci even after 3 days of IAA treatment (Fig. 7a), is reminiscent of a significant (27%) retention of CTCF-AID peaks that were reported after 2 days of exposure to IAA[34]. Since most other inducible loss-of-function approaches display lower efficiency and slower kinetics, it is likely that they may result in even slower protein depletion from chromatin.

The distinct roles of PRDM14 in mouse and human PGCs, further illustrate the impact of the species-specific regulatory networks for PGC fate. For example, expression of SOX17 and SOX2 is mutually exclusive in germ cells of human and mouse, respectively. Whereas SOX17 is essential for hPGC fate[3], SOX2 promotes mPGC specification and survival[49]. By contrast, repression of SOX2 is apparently prerequisite for the initiation of hPGCLC fate[1], since ectopic expression of SOX2 abrogates hPGCLC specification[50]. In the absence of BMP signalling, which initiates hPGCLC specification, high expression of SOX2 and PRDM14 is maintained (Supplementary Fig. 2). It is possible that PRDM14 expression in the hPGCLC-competent cells is downstream of SOX2. The repression of SOX2 upon the initiation of hPGCLC in response to BMP signalling might therefore explain a transient loss of PRDM14. Indeed, in hESC where SOX2 binds to PRDM14 promoter[51], loss of *SOX2* leads to *PRDM14* downregulation[52]. The observed re-expression of PRDM14 in hPGCLC follows after the upregulation of SOX17 and BLIMP1 at the onset of hPGCLC specification (Fig. 2 and Supplementary Fig. 2), which suggests a molecular shift in the regulation of PRDM14 expression in germ cells.

In mPGCs, PRDM14 promotes DNA demethylation, in part by repressing *Uhrf1*, and initiates the reduction of H3K9me2 through *Ehmt1* downregulation[9]. How the TF network might control the epigenetic resetting in hPGCs is less clear, but some essential features of epigenetic reprogramming are evident in hPGCs, including the repression of *UHRF1* and *EHMT2* (ref. [26]). While there was a trend towards derepression of UHRF1 in a subset of PRDM14-deficient hPGCLCs (Supplementary Fig. 8C), we did not observe changes in H3K9me2 levels. Unfortunately, currently available hPGCLCs only recapitulate pre-migratory hPGCs, and do not show significant epigenetic changes as observed in later-stage in vivo germ cells[1]. Therefore, it is difficult to comprehensively assess the influence of PRDM14 loss on epigenetic resetting in the human germline. Further studies are required to establish how precisely the epigenetic resetting is initiated in hPGCs and establish whether or not PRDM14 has an important role in this critical process.

Our study shows the importance of the use of rapid and comprehensive PRDM14 depletion for the phenotype unmasking. A previous study of PRDM14 in hPGCLCs used a partial knockdown at a later time-point (D2 BLIMP1$^+$SOX17$^+$ precursors), where the homogeneity and speed of PRDM14 depletion was not monitored[19]. We demonstrate the utility of AID and JAZ degrons to deplete endogenous proteins to study human cell fate decisions. Simultaneous use of both degrons allows independent control of two proteins, and the construction of degron switches as shown here (Fig. 5b, d). The knock-in of AID/JAZ together with a fluorescent reporter facilitates protein expression analysis and allows the use of the anti-GFP (or other epitope tags) antibody for IF and ChIP. Furthermore, tissue-specific TIR1 expression can allow spatial control over protein stability, as shown in *Caenorhabditis elegans* and *Drosophila melanogaster* without detectable side effects[53,54].

Neither mouse nor human adult tissues express PRDM14 except in some types of cancer[14,55], indicating the importance of studying the role of PRDM14 in normal embryogenesis. Inducible degrons offer a more precise control over proteins when studying the roles of critical TFs undergoing dynamic changes during normal and malignant development.

## Methods

**Human embryonic stem cell culture.** PGC-competent 4i hESCs (H9 (ref. [56]), WIS2-NANOS3-T2A-tdTomato (N3)[5] and cell lines derived from them) were cultured as in ref. [3] on irradiated mouse embryonic fibroblasts (MEFs) (GlobalStem) in 4i medium (Table 1). Media were replaced every day. hESCs were passaged by single-cell dissociation using 0.25% Trypsin-EDTA (GIBCO). 10 μM ROCK inhibitor (Y-27632, TOCRIS) was added for 24 h after passaging.

Conventional hESCs (used for hPGCLC differentiation via preME) were grown in Essential 8 (E8)[57] in plates pre-coated with 5 μg/ml vitronectin for at least 1 h. Media were replaced every day. Cells were passaged in clumps using 0.5 mM EDTA in PBS. All reagents for E8 hESC culture were from Thermo Fisher Scientific.

For transgene inductions in hESCs or during differentiation, 1 μg/ml doxycycline (dox, Sigma) and/or 0.5 μM Shield-1 (Clontech) were added to media, where specified. For depletion of AID-fused proteins, auxin (indole-3-acetic acid sodium salt, IAA, Sigma) was used at 100 μM (in H$_2$O) unless otherwise specified. For depletion of JAZ-fused proteins, coronatine (Cor, Sigma) was used at 50 μM (in DMSO) in all experiments. DMSO was used as vehicle control for experiments with Cor.

**HEK-293T cell manipulation.** HEK-293T cells were cultured in DMEM with 10% fetal bovine serum (FBS, Corning), 1 mM sodium pyruvate, 2 mM L-glutamine and PenStrep (all GIBCO)[24]. For JAZ degron testing, the cells were transfected with vectors for expression of GFP fused to *Os*JAZ[33], *Os*JAZ[33]-NLS or *Os*JAZ[43]. Lentivirus production, HEK-293T transfection and selection were carried out as in[24]. The cells were then treated with either 50 μM Cor (Sigma) in DMSO or 0.1% DMSO as control. GFP depletion was analysed using flow cytometry. FlowJo (TreeStar) was used to calculate the geometric mean of GFP fluorescence intensity.

**hPGCLC and DE induction.** To induce hPGCLCs[3], 4i hESCs were trypsinised, filtered and plated into ultra-low cell attachment U-bottom 96-well plates (Corning, 7007) at 4000 cells/well density in 100 μl PGCLC medium[5] (Table 2). For induction of large quantities of hPGCLCs (for ChIP-seq and qPCR experiments), six-well EZSPHERE microplates (ReproCELL) were used (500,000 cells/well in 3 mL PGCLC medium). The plates were centrifuged at 300 *g* for 3 min and placed into a 37 °C 5% CO$_2$ incubator until embryoid body (EB) collection for downstream analysis. Reporter fluorescence intensities were monitored daily throughout differentiation using Olympus IX71 microscope.

For hPGCLC induction from E8 (conventional) hESCs[5], preME competent state was induced by seeding 200,000 trypsinised single cells per well of a vitronectin-coated 12-well plate and culturing for 12 h in ME medium (Supplementary Table 1). PreME cells were then trypsinised, filtred and induced similar to 4i hESCs, as described above.

For DE induction[5], ME was first obtained from E8 cells by 36-h exposure to ME medium (Supplementary Table 1). ME was then washed with PBS followed by 48-h culture in the DE medium (Supplementary Table 2).

**Flow cytometry and fluorescence-activated cell sorting.** Flow cytometry and FACS were performed as in[3]. The gating strategy for hPGCLC identification is shown in Supplementary Fig. 10. At least six EBs were washed in PBS and dissociated with 0.25% Trypsin-EDTA for 10 min at 37 °C. Cells were washed, resuspended in FACS buffer (3% FBS in PBS) and incubated with anti-AP and anti-CD38 antibodies specified in Supplementary Table 3 for 30–60 min at 4 °C in the dark. After washing, the cells were resuspended in FACS buffer with 0.1 μg/ml DAPI and filtered through a 50 μm cell strainer. Flow cytometry was done using BD LSR Fortessa, while FACS was performed with Sony SH100 Cell Sorter. Data were analysed using FlowJo (Tree Star).

**Isolation of gonadal hPGC by FACS.** The collection and usage of human embryonic tissues were approved by the National Research Ethics Service (REC 96/085). Patients (who had already decided to undergo the termination of pregnancy operation) fully and freely consented to donate the foetal tissues for medical and academic research. Human genital ridges were dissected in PBS and separated from the mesonephros followed by dissociation with TrypLE Express (Life Technologies) at 37 °C for 20–40 min (depending on the tissue size) with pipetting every 5 min. Cells were washed, resuspended in FACS buffer (3% FBS and 5 mM EDTA in PBS) and incubated with anti-AP and anti-cKIT antibodies (specified in Supplementary Table 3) for 15 min at room temperature with 10 rpm rotation in the dark. Cells were then washed in FACS medium and filtered through a 35 μm cell strainer. FACS was performed with Sony SH100 Cell Sorter and data were analysed using FlowJo (Tree Star). Cell populations of interest were sorted onto Poly-L-Lysine Slides (Thermo Scientific) and fixed in 4% PFA for IF analysis.

**Immunofluorescence.** For IF, hESCs were grown on ibiTreat eight-Well μ-Slides (Ibidi) on MEFs (GlobalStem), washed with PBS and fixed with 4% paraformaldehyde (PFA, Thermo Fisher Scientific) in PBS at room temperature (RT) for

**Table 1 4i medium composition.**

| Component | Final concentration | Supplier |
|---|---|---|
| Knockout DMEM | – | GIBCO |
| Knockout serum replacement (KSR) | 20% | GIBCO |
| L-glutamine | 2 mM | GIBCO |
| Nonessential amino acids | 0.1 mM | GIBCO |
| 2-mercaptoethanol | 0.1 mM | GIBCO |
| Penicillin–streptomycin | 100 U/ml (penicillin) 0.1 mg/ml (streptomycin) | GIBCO |
| Human LIF | 20 ng/ml | Stem Cell Institute (SCI) |
| bFGF | 8 ng/ml | SCI |
| TGF-β1 | 1 ng/ml | Peprotech |
| CHIR99021 (CH) | 3 µM | Miltenyi Biotec |
| PD0325901 (PD) | 1 µM | Miltenyi Biotec |
| SB203580 (SB) | 5 µM | TOCRIS bioscience |
| SP600125 (SP) | 5 µM | TOCRIS bioscience |

**Table 2 hPGCLC medium composition.**

| Component | Final concentration | Supplier |
|---|---|---|
| Advanced RPMI 1640 | – | GIBCO |
| B27 supplement | 1% | GIBCO |
| L-glutamine | 2 mM | GIBCO |
| Nonessential amino acids | 0.1 mM | GIBCO |
| Penicillin–streptomycin | 100 U/ml (Penicillin) 0.1 mg/ml (Streptomycin) | GIBCO |
| BMP2 | 500 ng/ml | SCI |
| Human LIF | 1 µg/ml | SCI |
| SCF | 100 ng/ml | R&D systems |
| EGF | 50 ng/ml | R&D systems |
| ROCK inhibitor (Y-27632) | 10 µM | TOCRIS bioscience |
| Poly-vinyl alcohol (PVA, 10%) | 0.25% | Sigma |

10 min. This was followed by three washes in PBS and a 10-min permeabilisation in 0.25% Triton X-100 (Sigma) in PBS. Next, the samples were incubated with the blocking buffer (0.1% Triton X-100, 5% normal donkey serum (Stratech) and 1% bovine serum albumin (BSA, Sigma)) for 30 min at RT. The samples were then incubated with primary antibodies (Supplementary Table 3) diluted in the blocking buffer overnight at 4 °C. The following day they were washed 3 times with the wash buffer (0.1% Triton X-100 in PBS) and incubated with Alexa fluorophore (AF- 488, 568 and/or 647)-conjugated secondary antibodies (Invitrogen, used in 1:500 dilutions in the blocking buffer) specific for the host species of the primary antibodies for 1 h at RT in the dark. After three washes in the wash buffer, the samples were incubated in PBS with 1 µg/ml DAPI for 10 min at RT. The samples were stored at 4 °C in the dark (up to 2 weeks) before imaging on Leica SP5 inverted confocal microscope. The images were analysed using Fiji software.

hPGCLC-containing EBs were fixed on D4 (unless otherwise specified) using 4% PFA in PBS at 4 °C for 1–2 h. After two washes in PBS, the EBs were transferred to 10% sucrose solution in PBS and stored at 4 °C for 1 day. This was followed by a 1-day incubation in 20% sucrose in PBS at 4 °C. Finally, the EBs were embedded in OCT compound (CellPath), snap frozen on dry ice and stored at −80 °C until cryosectioning. Cryosections of 8-µm thickness were made on Superfrost Plus Micro slides (VWR) using a Leica Microsystems cryostat. Slides were stored at −80 °C or processed immediately. For IF, slides were air-dried at RT for 1 h, washed in PBS three times 5 min each and permeabilised in 0.1% Triton X-100 in PBS for 30 min at RT. The cryosections on each slide were then circumscribed using ImmEdge Hydrophobic Barrier Pen (Vector Labs) and blocking solution (as above) was added for 30 min at RT. Samples were then incubated with primary and secondary antibodies (Supplementary Table 3) as specified for hESC IF above, but 1 µg/ml DAPI was added to the secondary antibody mixtures. Finally, the slides were washed in the wash buffer twice and in PBS once and mounted with Prolong

Gold Antifade Reagent (Molecular Probes). The images were acquired using Leica SP5 confocal microscope and analysed using Fiji software[58].

A custom script for Fiji written by Dr Richard Butler (Gurdon Institute, University of Cambridge, UK) was used to segment nuclei with an area of 30–300 µm² in the DAPI channel and measure fluorescence intensity in other channels. The script is available at https://github.com/gurdon-institute/SOX17_PRDM14_Measurement. The data was then manually processed in Microsoft Excel to filter pluripotent cells (NANOG or OCT4 fluorescence intensity >50) where applicable and to calculate mean fluorescence intensities. Statistical analyses were performed using GraphPad Prism 7.

**Plasmid constructions and cell line establishment.** For gene targeting, CRISPR-Cas9 approach was used[25]. Annealed oligos (Supplementary Data 5) encoding corresponding gRNAs were ligated into pX330 vector[25] digested with BbsI. gRNA sequences were chosen using the MIT CRISPR tool (http://www.genome-engineering.org/crispr/) to be located close to the stop codon (to enable C-terminal fusions in PRDM14-T2A-Venus, PRDM14-AID-Venus, PRDM14-JAZ-Venus and SOX17-AID-Venus). The chosen protospacer sequences were as follows: GTGAAGACTACTAGCCCTGC for PRDM14 and GACGTGTGACAGGTCCC TGA for SOX17 (Naoko Irie, personal communication).

All other plasmids, including donor vectors for homology-directed repair were generated using In-Fusion cloning (Clontech) according to manufacturer's recommendations, but scaling down the reaction to a total volume of 5 µl. Primers used for cloning are specified in Supplementary Data 5.

For knock-ins either electroporation or lipofection was used. For electroporation, ~250,000 trypsinised hESCs were resuspended in 600 µl PBS + Ca²⁺ + Mg²⁺ containing 50 µg gRNA plasmid and 50 µg homologous repair donor plasmid and electroporated in a 0.4-cm cuvette using Gene Pulser Xcell System (Bio-Rad) with a single 20 ms square-wave pulse (250 V). Lipofections were performed using 2 µg gRNA plasmid and 2 µg donor construct in OptiMEM medium (GIBCO) and Lipofectamine Stem reagent (Invitrogen) according to manufacturer's recommendations. The volume of lipofectamine in microlitres was equal to the total amount of DNA in micrograms. For lipofection of PiggyBac transgenes a total of 1–5 µg DNA/100,000 cells was used, with the amount of PiggyBac transposase (PBase)-encoding plasmid equal to that of PiggyBac-delivered transgenes combined (in µg). Transfected cells were seeded onto drug-resistant DR4 MEFs (SCI) in 4i medium (ROCKi added for the first 24 h; selection was initiated 48 h after transfection).

After selection, individual clones were picked, expanded and genotyped using PCR. For this, genomic DNA was first extracted from cell pellets in the lysis buffer (10 mM Tris pH 8.0, 100 mM NaCl, 10 mM EDTA and 0.5% SDS; Proteinase K (PK) was added immediately prior to lysis at a final concentration of 0.2 mg/ml) at 56 °C for 4 h—overnight, followed by PK inactivation at 98 °C for 10 min. The supernatant was used directly in genotyping PCR performed with LongAmp (NEB) or PrimeStar GXL (Clontech) polymerase according to the manufacturers' protocols. Primers used for genotyping are specified in Supplementary Data 5. In the case of transgenes expression, homogeneity and leakiness of induction was checked by IF.

For knock-ins (PRDM14-T2A-Venus, PRDM14-AID-Venus, PRDM14-JAZ-Venus and SOX17-AID-Venus) correct targeting in homozygous (by genotyping) clones was confirmed by Sanger sequencing. This was followed by the removal of the Rox-flanked puromycin resistance cassette by transient transfection with 1 µg of Dre-recombinase-encoding plasmid[59]. Two days of hygromycin B selection (50 µg/ml) were followed by 2 days of negative selection using FIAU (200 nM)[28]. The obtained clones were again genotyped by PCR and sequenced to confirm antibiotic cassette excision. Venus intensity and homogeneity were then verified using Flow cytometry and IF.

For generation of cell lines that deplete PRDM14 upon IAA supplementation, AID-Venus was added at the C-terminus of PRDM14 using CRISPR (see above). This was followed by the addition of TIR1 transgene (pPB-CAG-OsTir1-myc-IRES-HygR or pPB-CAG-OsTir1-V5-T2A-Puro) by lipofection, along with pPBase. TIR1 cDNA and the 44-amino-acid AID sequence[34] were kindly provided by Dr Elphège Nora.

For generation of cell lines that deplete PRDM14 upon Cor supplementation, JAZ-Venus was added at the C-terminus of PRDM14 using CRISPR (see above). This was followed by the addition of COI1B transgene (pPB-CAG-HA-FboxOsTir1-OsCoi1b-IRES-HygR) by lipofection, along with pPBase.

For PRDM14-DD rescue, PRDM14-AID-Venus + TIR1 hESCs were co-lipofected with 0.05 µg pPB-CAG-myc-PRDM14-DD-IRES-HygR (low DNA amount was used to limit transgene copy number), 0.5 µg pPB-CAG-OsTir1-V5-T2A-Puro (to avoid TIR1 excision by PBase) and 0.55 µg PBase.

**Reverse-transcription quantitative PCR (qPCR).** Total RNA was extracted using the RNeasy Mini Kit (QIAGEN) from unsorted cells or using Arcturus PicoPure (Thermo Fisher Scientific) from at least 1000 sorted cells. cDNA was synthesised using the QuantiTect Reverse-Transcription Kit (QIAGEN). qPCR was performed on a QuantStudio 12K Flex Real-Time PCR machine (Applied Biosystems) using SYBR Green JumpStart Taq ReadyMix (Sigma) and specific primers (Supple-mentary Data 5). The ΔΔCt method was used for quantification of gene expression.

Statistical analyses were performed using Microsoft Excel and/or GraphPad Prism 7.

**RNA-sequencing**. RNA-seq was performed on PRDM14-AID-Venus competent 4i hESCs and hPGCLCs induced therefrom. Two biological replicates were used for each condition. For IAA-sensitive cells, two clones (cl11 and cl21) from the same hESC passage or the same hPGCLC induction were used as replicates. For the parental (no TIR1) control, the same cell line was used at different passages or inductions to yield two independent replicates. In total, 10,000 AP$^+$ 4i hESCs or 10,000 NANOS3-tdTomato$^+$AP$^+$ hPGCLCs (with the exception of hPGCLC cl21 replicate, where 3000 cells were used) were sorted directly into 100 μl of extraction buffer from the PicoPure RNA Isolation Kit (Applied Biosystems) for subsequent total RNA extraction according to manufacturer's protocol. RNA was stored at −80 °C and its quality and quantity were checked by the Agilent RNA 6000 Pico Kit with Bioanalyzer (Agilent Technologies) and Qubit (Thermo Fisher Scientific). RNA-seq library was prepared from 10 ng input RNA using the end-to-end Trio RNA-seq library prep kit (Nugen) following the manufacturer's protocol but omitting the AnyDeplete step. In short, the protocol contains the following steps: DNAse treatment to remove DNA from RNA; first strand and second strand cDNA synthesis to produce the reverse complement of the input RNAs; cDNA purification using Agencourt AMPure XP beads (Beckman Coulter); single-primer isothermal amplification to stoichiometrically amplify cDNAs; enzymatic fragmentation and end repair; sequencing adaptor (index) ligation; product purification using AMPure beads; library amplification (four cycles were used); library purification using AMPure beads. Libraries were then quantified by qPCR using the NEBNExt Library Quant Kit (NEB) for Illumina on QuantStudio 6 Flex Real-Time PCR System (Applied Biosystems). Fragment size distribution and the absence of adaptor dimers was checked using Agilent TapeStation 2200 and High Sensitivity D1000 ScreenTape. Finally, RNA-seq libraries were subjected to single-end 50 bp sequencing on HiSeq 4000 sequencing system (Illumina). Twenty-four indexed libraries (including samples from this work and eight others) were multiplexed together and sequenced in two lanes of a flowcell, resulting in >30 million reads per sample.

**RNA-seq analysis**. The library sequence quality in demultiplexed fastq files was checked by FastQC (v0.11.5)[60] and the low-quality reads and adaptor sequences were removed by Trim Galore (v0.4.1)[61] using the default parameters. The pre-processed RNA-seq reads were mapped to the human reference genome (UCSC GRCh38/hg38) using STAR (2.6.0a)[62] (–outFilterMismatchNoverLmax 0.05 –outMultimapperOrder Random –winAnchorMultimapNmax 100 –outFilterMultimapNmax 100) guided by the ENSEMBL (Release 87) gene models. Read counts per gene were extracted using TEtranscripts[63] and normalised by DEseq2 in R[64]. Differential expression analysis was also performed using DEseq2. The resulting gene expression table (Supplementary Data 1) was used for downstream analyses in Microsoft Excel and R. Pearson's correlation analysis was performed using the R cor command. Unsupervised hierarchical clustering was performed using the R hclust function with (1—Pearson's correlation coefficient) as distance measures. The R prcomp function was used for PCA. GSEA was performed using the GSEA software by the Broad Institute[65]. GO analysis was performed using DAVID[66].

For comparative differential expression analysis of PRDM14 depletion in hPGCLCs and TFAP2C and PRDM1 KOs from[20] the samples were identically pre-processed and mapped to the human reference genome (UCSC GRCh38/hg38), sequencing reads were re-normalised using DEseq2 to generate a new integrated gene expression table (Supplementary Data 2). A similar procedure was performed for comparisons with mouse Prdm14 KO from[11], where processed reads were mapped to the mouse reference genome (UCSC GRCm38/mm10), and high-confidence human–mouse one-to-one orthology assignments for protein-coding genes were obtained from ENSEMBL (Release 87) (Supplementary Data 4). Venn diagrams were plotted using VennPainter[67] and p values for two overlapping datasets were calculated using a generalised hypergeometric test for multiple samples[68]. For the co-expression analysis of DEGs, the gene-based Pearson's correlation coefficients were calculated for PRDM14-AID-Venus competent hESC and hPGCLC samples using R. All pairwise correlations $r < 0.8$ between genes were removed. The matrix was imported as an adjacency matrix into the R igraph package[69], and the Louvain method for community detection was performed on the resulting graph. For comparison with microarray data[13], the published data was normalised, log2-transformed and the differential expression was evaluated using the R limma package[70].

**Chromatin immunoprecipitation (ChIP) and ChIP-seq**. ChIP was performed using the SimpleChIP Enzymatic Chromatin IP Kit (with Magnetic Beads) from Cell Signalling Technology (CST) following manufacturer's recommendations with modifications. Recipes for all buffers are provided in Table 3. The following samples were included in the analysis (one replicate per condition): 4i hESCs (cl11 no IAA), 4i + IAA hESCs (cl11 + IAA) and hPGCLCs (cl11 and cl21 pooled; no IAA). For each ChIP 2.5 million unsorted 4i hESCs or 4 million unsorted hPGCLCs (D4 EBs containing 60–70% PGCLCs as assessed by flow cytometry) were used. The cells were washed in PBS, filtered through 50 μm strainer and fixed

**Table 3 Buffer recipes for ChIP.**

| | |
|---|---|
| Buffer A | 1x |
| Buffer A stock, μl | 250 |
| H$_2$O, μl | 750 |
| DTT, μl | 0.5 |
| PIC, μl | 5 |
| Buffer B | 1x |
| Buffer B stock, μl | 275 |
| H$_2$O, μl | 825 |
| DTT, μl | 0.55 |
| ChIP buffer 1 | 1x |
| ChIP buffer stock, μl | 10 |
| H$_2$O, μl | 90 |
| PIC, μl | 0.5 |
| ChIP buffer 2 | 1x |
| ChIP buffer stock, μl | 40 |
| H$_2$O, μl | 360 |
| PIC, μl | 2 |
| tRNA (Sigma) [10 mg/ml], μl | 2 |
| Elution | 1x |
| Elute buffer, μl | 25 |
| H$_2$O, μl | 25 |
| Blocking buffer | 1x |
| ChIP buffer (CST), μl | 150 |
| H$_2$O, μl | 1110 |
| PIC (Roche), μl | 75 |
| BSA (Sigma) (10%), μl | 150 |
| tRNA (Sigma) [10 mg/ml], μl | 15 |
| Low salt | 1x |
| ChIP buffer (CST), μl | 300 |
| H$_2$O, μl | 2700 |
| High salt | 1x |
| ChIP buffer (CST), μl | 100 |
| H$_2$O, μl | 900 |
| NaCl, μl | 70 |
| LiCl Wash buffer | For 250 mL |
| 20 mM Tris-HCl pH 8.0, ml | 5 ml of 1M |
| 2 mM EDTA, ml | 1 ml of 0.5 M |
| 250 mM LiCl, g | 2.65 g |
| 1% NP-40, ml | 2.5 ml |
| 1% Deoxycholate, g | 2.5 g |
| H2O, ml | 241.5 ml |

in 1% formaldehyde at RT for 10 min. Formaldehyde was quenched by adding 450 μl of 10× glycine at RT for 10 min, followed by centrifugation at 500 g, 4 °C for 5 min. The cells were then washed twice in ice-cold PBS with protease inhibitor cocktail (PIC) and the pellets were snap frozen on dry ice and stored at −80 °C. To prepare nuclei, cells were thawed on ice, washed in ice-cold buffers A and B, resuspended in 100 μl buffer B and transferred to a thin-walled Diagenode 1.5 ml tubes. Next, 0.35 μl micrococcal nuclease (MNase) was added and the suspension was incubated for 23 min at 37 °C, shaking at 800 rpm to enzymatically digest the chromatin. The addition of 10 μl 0.5 M EDTA for 2 min on ice was used to stop the digestion. Next, the nuclei were pelleted at 15,500 g, 4 °C for 1 min, resuspended in 100 μl of ChIP buffer 1 and kept on ice for 10 min, followed by sonication (3 × 30 s on/off, high output) using Bioruptor sonicator (Diagenode). The lysate was cleared by centrifugation at 9200 g, 4 °C for 10 min and transferred to a fresh DNA LoBind 1.5 ml tube (Eppendorf). For ChIP, 400 μl ChIP buffer two was added for a total of 500 μl/sample; 10 μl were removed and stored at −80 °C to serve as 2% input. Antibodies (3 μl PRDM14 ABD121 or 0.5 μl GFP ab290, see Supplementary Table 3) were then added to each immunoprecipitation (IP) reaction and IPs were incubated overnight rotating at 10 rpm, 4 °C.

Equal volumes of Protein A and Protein G-conjugated magnetic beads (Invitrogen) were combined (for a total of 20 μl per IP) in a DNA LoBind 1.5 ml tube, washed in 200 μl blocking buffer, and incubated with 500 μl blocking buffer for 1 h rotating at 10 rpm, 4 °C. The tubes were placed on a magnetic rack (Diagenode) to remove the supernatant and another 1-h incubation was performed. Then the supernatant was removed, the beads were resuspended in 20 μl blocking buffer and stored at 4 °C for up to 24 h until the use in ChIP.

After overnight IP, 20 μl of pre-blocked Protein A/G beads were added to each sample, followed by rotation at 10 rpm, 4 °C for 2 h. The tubes were placed on a magnetic rack and the supernatant was removed. The beads were then washed three times with 1 ml low-salt buffer (each wash involved a 5-min rotation at 10 rpm, 4 °C),

once with high-salt buffer and once with LiCl buffer. Finally, the beads were resuspended in 50 μl elution buffer and transferred to 0.2 ml PCR stripes. The inputs were thawed on ice and supplemented with 50 μl elution buffer. All samples were incubated at 65 °C, 110 g for 30 min to elute the protein–DNA complexes from the beads. The stripe was then placed onto a magnetic rack and the eluate was transferred into a fresh 0.2 ml PCR stripe to which 2 μl 5 M NaCl and 2 μl PK (10 mg/ml) were added. The samples were then incubated at 65 °C, 110 g for 2 h for decrosslinking.

Finally, for the use in ChIP-qPCR, DNA was purified using MinElute columns (QIAGEN) following the manufacturer's protocol. Alternatively, for ChIP-seq library preparation, DNA was purified using Agencourt AMPure XP beads (Beckman Coulter) as follows. 1.8× volume of AMPure beads were mixed with each sample, incubated for 10 min to bind DNA to the beads and washed 2 × 30 s with 200 μl 80% ethanol using a magnetic rack. The beads were air-dried for 5–10 min to reach complete ethanol evaporation. Thirty three microlitres of elution buffer (10 mM Tris-HCl, pH 8.0–8.5) was added and incubated with the beads for 2 min to elute the ChIP DNA. The PCR stripe was placed on the magnetic rack and 30 μl of eluate was transferred into fresh tubes.

For ChIP-seq libraries were prepared using the KAPA HyperPrep Kit following the manufacturer's instructions. Briefly, the protocol contains the following steps: end-repair and A-tailing; sequencing adaptor (index) ligation; product purification using AMPure beads; library amplification using the KAPA real-time Library Amplification Kit (11 cycles were used for ChIP libraries and 7 for input libraries to achieve similar concentration range); product purification using AMPure beads. Libraries were then quantified by qPCR using the NEBNExt Library Quant Kit for Illumina (NEB) on QuantStudio 6 Flex Real-Time PCR System (Applied Biosystems). Fragment size distribution and the absence of adaptor dimers was checked using Agilent TapeStation 2200 and High Sensitivity D1000 ScreenTape. Finally, ChIP-seq libraries were subjected to single-end 50 bp sequencing on HiSeq 4000 sequencing system (Illumina). Eight indexed libraries were multiplexed in one lane of a flowcell, resulting in >20 million reads per sample.

**ChIP-seq analysis**. The library sequence quality in demultiplexed fastq files was checked by FastQC (v0.11.5)[60] and the low-quality reads and adaptor sequences were removed by Trim Galore (v0.4.1)[61] using the default parameters. The trimmed ChIP-seq reads were aligned to the human reference genome (UCSC hg38) by the Burrows–Wheeler Aligner (0.7.15-r1142-dirty)[71]. Samtools (version: 1.3.1)[72] was used to remove unmapped and low-mapping quality reads (options: view -F 4 -q 20). Subsequently, duplicated reads and reads mapped to unlocalised contig (random), unplaced contigs (ChrUn) and blacklisted regions (obtained from http://mitra. stanford.edu/kundaje/akundaje/release/blacklists/hg38-human/) were removed using Samtools rmdup. This was followed by preliminary peak calling using macs2 call-peak[73] against the corresponding inputs (options: -g 3e9 –keep-dup all). Percentage of reads in peaks was calculated using featureCounts[74] and visualised with multiQC[75]. Before peak calling, to compare the number of peaks between the three high-quality samples, the library size for all libraries was downsampled to 25 million reads. Peaks were then called again using macs2 callpeak against the corresponding inputs (options: -g 3e9 –keep-dup all –nomodel –extsize 157). The resultant peak files of the three samples were merged to generate a consensus set of 6486 peaks using bedtools merge (options: -d 100)[76]. The consensus peaks were processed and clustered by DeepTools computeMatrix and plotHeatmap[77]. Peak annotation, motif enrichment and differential peak analyses were performed by HOMER[41]. DeepTools bam-Coverage (options: -extendReads 157 –binSize 10 –normalizeUsing CPM –ignoreForNormalization chrX chrY) was used to generate bigwig file for visualisation in Integrative Genomics Viewer[78]. The integrated ChIP-seq and RNA-seq table (Supplementary Data 3) was generated using R to correlate changes in gene expression to PRDM14 binding. Venn diagrams were plotted using the R/Bioconductor ChIPseeker package[79].

**Reporting summary**. Further information on research design is available in the Nature Research Reporting Summary linked to this article.

## Data availability

All relevant data, including full plasmid sequences, are available from the authors on request. The source data underlying experimental results for Fig. 1d is provided as a Source Data file. High-throughput sequencing (RNA-seq and ChIP-seq) data have been deposited at the NCBI Gene Expression Omnibus (GEO) database and can be found at the following address: https://www.ncbi.nlm.nih.gov/geo/query/acc.cgi?acc=GSE138675.

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

## Acknowledgements

We thank Dr Elphège Nora for kindly providing reagents (plasmids containing AID degron and TIR1 sequences) and guidance for the establishment of the auxin-inducible degron. We are grateful to Dr Richard Butler for writing the Fiji plugin for fluorescence intensity quantification. We gratefully acknowledge Dr Toshihiro Kobayashi, Dr Naoko Irie and Dr Ufuk Günesdogan for their useful advice throughout the project. A.S. was funded by the 4-year Wellcome Trust PhD Scholarship; M.P.S. was funded by a Churchill scholarship; MAS holds a Wellcome Senior Investigator Award, and grant from the MRC. The Gurdon Institute is supported through core funding from the Wellcome Trust and Cancer Research UK. W.H.G. was supported by an EMBO long term fellowship (ALTF 263-2014). W.W.C.T. holds a Croucher Fellowship for Postdoctoral Research.

## Author contributions

The study was conceived and designed by A.S. and M.A.S. A.S. performed most experiments; W.W.C.T. and S.D. performed most bioinformatics analyses; W.W.C.T. prepared human gonadal sections and conducted some IF stainings of human gonadal sections. M.P.S helped with revision experiments, including time-course hPGCLC induction for some IF experiments and time-course of TIR1-JAZ rescue. W.H.G. helped with ChIP experiments and data analysis. R.B. provided reagents for the JAZ degron and contributed JAZ degron sequence characterisation in HEK-293T cells. The study was supervised by M.A.S. The paper was written by A.S. and M.A.S. with contributions from most authors.

## Competing interests

The authors declare no competing interests.
