## [Peer Review File · Nature Communications]

Reviewers' comments:

Reviewer #1 (Remarks to the Author):

RPDM14 is involved in the maintenance of hESC and mESC and is known to be important for PGC specification in mice. To understand the importance of RPDM14 for hPGCLC specification and the differences in its role between human and mice, Sybrina et al. applied AID and JAZ degrons to deplete RPDM14 in hESC. The authors showed that RPDM14 depletion was achieved upon the addition of the inducing ligand (IAA/Cor) and showed that AID worked more rapidly and efficiently than JAZ (Figure 3). Timely depletion of RPDM14 significantly reduced hPGCLC induction efficiency, clearly showing that the importance of RPDM14 during hPGCLC specification (Figure 4). They used multiple rescue assays to test the AID system and validated the importance of RPDM14 for hPGCLC specification (Figure 5). Transcriptome and ChIP-seq analyses with hESC and hPGCLC depleted of RPDM14 strongly suggested that RPDM14 functions together with TFAP2C and BLIMP1, and that the target genes controlled by RPDM14 in humans are significantly different from those in mice (Figure 6 and 7).

This is a technically sound and beautiful paper. The authors successfully established the AID and JAZ degron systems in hESC. They nicely performed multiple rescue experiments to validate the AID system. Because target depletion using AID is quicker and more efficient than that using conventional conditional systems (e.g. siRNA), the effect to hPGCLC specification and transcriptome was clear and more pronounced. This is a high-quality paper and I do not see a major problem for publication.

Minor points

This paper emphasizes the advantages of new genetic systems. However, I could not find the details of some important methodologies in Materials and Methods.

1. The gRNA sequences of CRISPR-Cas9 should be shown.
2. The JAZ system has been less commonly used since it was originally established (Brosh et al. Nature Communications, 2016). In the original publication, they tested multiple COI1B variants and JAZ degrons. It is not clear for me which combination was used. The authors should describe the details of the used JAZ system. It might be worth considering to show the sequence information of key components.

Reviewer #2 (Remarks to the Author):

Molecular mechanisms of germ cell specification in human as well as in mouse have been revealed. Although Prdm14 is crucial in mouse germ cell formation, previous in vitro knock-down (KD) experiments suggested that PRDM14 was dispensable in specification of human germ cells. Because the KD experiments likely resulted in insufficient loss of function of PRDM14 in human primordial germ cell (PGC)-like cells (PGCLCs), the authors re-evaluated possible roles of PRDM14 by the elegant inducible degron strategy which results in quick and efficient loss of the PRDM14 function.

The authors first established the mVenus knock-in cell lines to visualize the expression of the endogenous PRDM14 in the course of PGCLC development from human ES cells (hESCs) in culture. As a result, the authors found that transient initial downregulation and subsequent upregulation of nuclear PRDM14 during PGCLC induction. The authors then applied two different degron, i.e., AID-IAA-TIR1 and JAZ-Cor-COI1B systems. After addition of IAA, PRDM14 was diminished to nearly control levels within 10 min in hESCs. Cor was less effective compared with IAA, but PRDM14 was also declined. During PGCLC induction, ratios of PGCLCs were significantly reduced by IAA or Cor, and elimination of TIR1 by using TIR1-JAZ transgenes rescued IAA induced reduction of PGCLCs. In addition, induced stabilization of PRDM14 transgene product by the ProteoTuner system also rescued

IAA-induced PGCLC reduction. Transcriptome analysis revealed downregulation of hPGC- and pluripotency-related genes in PRDM14-depleted PGCLCs, and ChIP-seq confirmed the enrichment of PRDM14 in those genes, suggesting that PRDM14 target them. In addition, up- or down-regulated genes among PRDM14-, BLIMP1-, and TFAP2C-deficient hPGCLCs were significantly overlapped, and ChIP-seq showed the enrichment of PRDM14 in binding motifs of TFAP2C, BLIMP1 and SOX15 as well as of PRDM14 itself, suggesting that those transcription factors cooperatively control their target genes. The authors finally indicated that overlap of up- or down-regulated genes in PRDM14-deficient human and mouse PGCLCs were minimal, suggesting distinct functions of human and mouse PRDM14 in human and mouse.

This is an interesting and elegant study showing crucial functions of human PRDM14 in PGCLCs by using degron, and provide significant information to understand mechanisms of PGC specification. The manuscript is well-written as a whole. I would suggest a few minor comments described below.

Specific comments;

1. More detailed descriptions concerning generating knock-in (KI) of AID (JAZ)-mVenus-PGK-puro in PRDM14 locus by CRISPR-Cas9 are helpful to follow this KI strategy.
2. Fig. 5A, page 13: It is unclear why stabilization of PRDM14 at day 1 was not so effective as that at day 0 to rescue decreased PGCLC by IAA.
3. Fig. 6G, Suppl. Fig. 5G, Suppl. Fig. 6B: GO analysis for down-regulated genes by PRDM14 depletion in hPGCLCs may be informative.
4. Page 26, the last two lines, 'after the upregulation of SOX17 and BLIMP1 at the onset of hPGCLC specification (Fig.2 and Suppl. Fig.2)': Those figures do not show the expression of BLIMP1, but Suppl. Fig.2 show the expression of TFAP2C.
5. Page 3, the last line: 100 μ M may be 100 μ m.

Reviewer #3 (Remarks to the Author):

In this new paper from the Surani laboratory, the authors used a novel degron approach to examine the role of PRDM14 in hPGC development using the differentiation of hPGCLCs from pluripotent stem cells as a model. This study is important because in the mouse, PRDM14 is required for hPGCLC specification and in a previous study a knockdown of PRDM14 had no effect on hPGCLC formation. This apparent species-specific difference is in line with the argument that new strategies have evolved to regulate hPGCLC formation (for example SOX17). However, in the first study the knockdown was very inefficient, leading to the possibility that the lack of phenotype could have been technical rather than biological. Here, the Surani lab have performed a comprehensive study that suggests PRDM14 is indeed involved in hPGC formation. Overall, this is a well-conceived study with interesting results that point to PRDM14's role in hPGCLC differentiation but with different targets. One of the major criticisms is that some of the findings are based on the analysis of what appears to be n=1 replicate or in some cases n=1 cell. The work would be better served by additional replicates to confirm key findings as indicated below.

For Sup fig 1A the authors claim both nuclear & cytoplasmic staining for PRDM14 protein. It would be helpful to see OCT4 and SOX17 merged with PRDM14 to confirm the intracellular localization. Furthermore, the legend indicates that both samples are female, but the figure is labelled 8wk male and 7wk female. Which is correct? If the figure label is correct, this would imply that the 7 wk female has nuclear PRDM14 in the hPGCs and the 8 wk male has cytoplasmic PRDM14 staining. For the 9 wk female sorted hPGCs, PRDM14 was nuclear. Given this apparent dynamic and possibly sex-specific localization of PRDM14 samples, more replicates around these time points from both male and female

embryos are needed before drawing conclusions regarding PRDM14 protein in hPGCs.

For Fig 2: It was unclear whether the lack of PRDM14 in SOX17 cells at 12 hr is because the SOX17 positive cells are endodermal (not germ cells) or as the authors suggest that PRDM14 is not required at this stage of hPGCLC formation. To address this the authors should stain with a second hPGC marker to determine whether the SOX17+ cells are indeed hPGCLCs at 12 hours. We see in the supplement there is also an image showing TFAP2C +/SOX2- PRDM14+ cell at 12hr, but a similar argument can also be made for the possibly the TFAP2C single positive cell is also not a hPGCLCs. Please label the images with +Cyto (as was done for no cyto) to make the interpretation clearer.

For Fig4. B&C. The authors have indicated based on previous studies that PRDM14 is required for pluripotent self-renewal, therefore inducing a knockout in PRDM14 at D0, followed by a loss of hPGCLCs may be due to either a direct role for PRDM14 in hPGCLC specification, or an indirect role related to the general loss of pluripotency and propensity for somatic cell differentiation. Given uncoupling this result (direct versus indirect) is central to determining the role of PRDM14 in hPGCLC specification, the authors need to show more specifically in the earliest stages of PRDM14KO that either the somatic cell differentiation is unaffected, or if it is affected, then whether this does or does not have an impact on hPGCLC specification. Particularly, given that removing PRDM14 in specified hPGCLCs at day 4 has not affect on hPGCLCs.

In the text and in Sup Fig 4, the authors state that PRDM14 depletion in 4i for one passage had no effect on hPGCLC specification. If PRDM14 is essential for human pluripotent stem cell self renewal, how did the cells remain pluripotent without PRDM14? Is PRDM14 not needed when cells are maintained in 4i conditions versus regular primed?

In Fig 5. D the authors performed a rescue with a shield which was successful on D0, but not successful at D1. Given this timing, what are the authors claiming? I am confused.

The authors performed RNA-Seq and show in Figure 6 NANOG down regulation. However in Figure 3 NANOG protein is still expressed in PRDM14 depleted cells. Which is correct? Furthermore, analysis of the 2015 Sasaki et al., Cell Stem Cell paper on hPGCLC differentiation from primed hESCs shows that hPGCLCs have background levels of PRDM14 RNA. How can these different RNA-seqs be reconciled? Is Prdm14 RNA uniquely regulated in human pluripotent stem cells and the earliest stages of human germ cell differentiation thus leading to different results on different platforms? The expression of Prdm14 RNA needs to be better described.

It appears that one of the P14-AID lines in Figure 6 is quite different in the PCA. Given that these are not true biological replicates but rather N=1 for two different cell lines it will be important to repeat these experiments from at least 1-2 more replicates from each cell line.

The claim that PRDM14 potentially plays a role in UHRF1 in hPGCLCs, similar to mice, would be strengthened if they quantified how often Ki67+ UHRF+ SOX17+ cells occurred. Not convinced by showing one cell that fits this criterion.

In the discussion, the authors claim that PRDM14 is likely downstream of SOX2, however, this would be contradictory to the results of the RNA seq, as a KO of PRDM14 in hESCs leads to increased SOX2 expression. Are the authors suggesting that PRDM14 is a negative regulator? Next, the hypothesis is that PRDM14 is upregulated after expression of SOX17 and BLIMP 1 (ref fig 2, sup fig 2), but BLIMP was never evaluated so how can this claim be made? In addition, the authors discuss PRDM14 -def hPGCLCs as exhibiting de-repressed UHRF1, but there is just one cell in an IF image to prove this,

and given the importance of this potential result, stronger evidence is needed to support this claim. Another statement in the discussion is the claim that PRDM14 mutant hPGCLCS do not de-repress EHMT2, but there was no evidence for this statement in this paper, the first time they mention EHMT2 is the discussion, there is no H3K9me2 staining.

RESPONSE TO REFEREES LETTER

We are grateful to the reviewers for their thoughtful and constructive comments. After carefully addressing the reviewers' comments we have incorporated additional experiments and replicates, which strengthen the conclusions on the critical role of PRDM14 in human PGC specification.

In particular, we have added new experimental data and/or additional replicates as follows:

- Additional IF analyses of male and female gonadal sections to address PRDM14 subcellular localisation in hPGCs (Reviewer #3)
- Comparison of GFP degradation in HEK-293 cells using different JAZ degron sequences (to substantiate our choice of specific JAZ degron reagents, Reviewer #1)
- The efficiency of GFP degradation in hESCs using the JAZ degron system (to substantiate our choice of specific JAZ degron reagents, Reviewer #1)
- Time-course IF analysis to assess the expression of PRDM14, SOX17, BLIMP1, OCT4 and KLF4 during hPGCLC specification, which confirms BLIMP1 upregulation prior to PRDM14, Reviewers #2 and #3; to test the specificity of other hPGCLC markers during hPGCLC induction and their expression dynamics relative to PRDM14, Reviewer #3)
- Time-course rescue of hPGCLC specification using the double degron (PRDM14-AID/TIR1-JAZ) approach (to test if it yields similar results as the time-course rescue using PRDM14-DD overexpression, suggesting that PRDM14 is important on D0 of specification, Reviewers #2 and #3)
- Additional IF replicates of UHRF1 expression in hPGCLCs and soma (to quantify the effect of PRDM14 loss on potential UHRF1 derepression in proliferating hPGCLCs, Reviewer #3)
- IF analysis of H3K9me2 levels in hPGCLCs and soma (to quantify the effect of PRDM14 loss on potential changes in H3K9me2 levels in proliferating hPGCLCs, Reviewer #3)
- qPCR to assess the expression of lineage markers in soma of D2 EBs induced with or without PRDM14 depletion (to check if PRDM14 loss does not affect somatic differentiation, Reviewer #3)
- IF analysis of endoderm markers expression in soma of D4 EBs induced with or without PRDM14 depletion (to check if PRDM14 loss does not affect somatic differentiation, Reviewer #3)
- A heatmap of expression of PRDM family members in hESCs, competent pre-mesendoderm and hPGCLCs induced therefrom (to show that PRDM14 is specifically expressed in hPGCLCs irrespective of the method of induction of competent state, Reviewer #3)

We have also added clarifications and additional experimental details, as requested by Reviewers #1 and #2.

All new figures in the Rebuttal are shown below; their inclusion in the revised manuscript is indicated where applicable

Reviewer #1 (Remarks to the Author):

This is a technically sound and beautiful paper. The authors successfully established the AID and JAZ degron systems in hESC. They nicely performed multiple rescue experiments to validate the AID system. Because target depletion using AID is quicker and more efficient than that using conventional conditional systems (e.g. siRNA), the effect to hPGCLC specification and transcriptome was clear and more pronounced. This is a high-quality paper and I do not see a major problem for publication.

Reviewer's supportive comments on the use of degrons to address the role of PRDM14 in hPGCLCs are much appreciated.

Minor points

This paper emphasizes the advantages of new genetic systems. However, I could not find the details of some important methodologies in Materials and Methods.

We have added extensive descriptions of the new genetic systems as described below:

1. The gRNA sequences of CRISPR-Cas9 should be shown.

The sequences of the gRNA oligos were in the Supplementary Table 6, and now also in the Materials and Methods.

2. The JAZ system has been less commonly used since it was originally established (Brosh et al. Nature Communications, 2016). In the original publication, they tested multiple COI1B variants and JAZ degrons. It is not clear for me which combination was used. The authors should describe the details of the used JAZ system. It might be worth considering to show the sequence information of key components.

We provide additional information on JAZ degron in both Results and Methods. We clarify the combination of hormone receptor and degron sequence we used (see below) and provide additional data on the selection of the JAZ degron sequence (Suppl. Fig.3 in the revised manuscript and Rebuttal Figure 1).

Based on optimisations of both hormone receptor and the degron sequence, the previous study (Brosh et al. 2016) showed highest efficiency of target protein degradation by JAZ degron under the following conditions: 1) When *OsCOI1B* hormone receptor is used (isoform B binds to a larger number of JAZ proteins; receptor from rice is better suited for use in mammalian cells than that from *Arabidopsis*, presumably because it functions better at 37 °C); 2) *OsCOI1B* is fused with *OsTIR1* F-box domain (ensures better integration into human SCF (SKP1, CUL1 and F-box) E3 ubiquitin–ligase complex); 3) the target protein is fused to *OsJAZ*³³ degron sequence (functions better than the equivalent sequence from *Arabidopsis* or the shorter *OsJAZ*²³); 4) the target protein is nuclear or targeted to the nucleus by a nuclear localisation signal (NLS);

We have since compared the best performing degron sequences from (Brosh et al. 2016), namely *OsJAZ*³³ (version 6-*Os33*) and NLS-*OsJAZ*³³ (version 7), with a slightly longer *OsJAZ*⁴³ degron sequence (without NLS). For this, HEK-293T cells were infected with viral constructs harbouring GFP fused to respective degron sequences and treated with coronatine (Cor) or DMSO (control). GFP depletion at 2, 4 and 24 hours was monitored by flow cytometry (Rebuttal Figure 1A). This showed that the longer degron sequence (*OsJAZ*⁴³) is superior to *OsJAZ*³³ and even slightly better than NLS-*OsJAZ*³³ in the efficiency and speed of GFP depletion (Rebuttal Figure 1A). This suggests that *OsJAZ*⁴³ is more versatile and can allow depletion of both nuclear and cytoplasmic proteins.

To further test if JAZ could be efficiently used in our system, we generated hESCs expressing a PiggyBac-delivered construct where GFP was fused to the best *OsJAZ*⁴³ degron sequence and an NLS, to better model the depletion of transcription factors such as PRDM14 (Rebuttal Figure 1B). The construct also harboured the Fb-TIR1-COI1B hormone receptor that renders the cells responsive to Cor. GFP signal started decreasing 3 hours after Cor administration (Rebuttal Figure 1C) and 99% of cells were GFP-negative after one passage in Cor (Rebuttal Figure 1D). Based on these results, we used the *OsJAZ*⁴³ and Fb-TIR1-COI1B combination to deplete PRDM14 (Fig.3C) or TIR1 (Fig.5).

The sequences of all constructs used in the study (for AID, JAZ and rescue experiments) have been uploaded as Related Manuscript Files.

Rebuttal Figure 1. Selection of JAZ degron sequence and its validation in hESCs. (A) GFP depletion in HEK-293-T cells by three indicated JAZ degron versions. GFP fluorescence intensity was measured by flow cytometry after indicated times of Cor or DMSO (control) treatment. Data is from n=1-2 independent experiments and is shown as relative geometric mean fluorescence intensity (normalised to DMSO control). (B) Scheme of the construct used to express GFP-JAZ in hESCs. COI1B (coronatine receptor) from rice (*Oryza sativa*) was fused to the F-box domain of TIR1 (Fb-TIR1) as in (Brosh et al. 2016). GFP fluorescent protein was fused to the JAZ degron. P2A and T2A – self-cleaving peptides. NLS – nuclear localisation signal. Puro – puromycin resistance gene. (C) Epifluorescence and bright-field photographs of GFP-JAZ hESCs after 3 hours or 1 day of Cor treatment. DMSO was used as a negative control for GFP depletion. (D) Flow cytometry plots confirming GFP-JAZ depletion after one passage in the presence of Cor.

Reviewer #2 (Remarks to the Author):

This is an interesting and elegant study showing crucial functions of human PRDM14 in PGCLCs by using degron, and provide significant information to understand mechanisms of PGC specification. The manuscript is well-written as a whole. I would suggest a few minor comments described below.

We thank the Reviewer for the positive comments.

Specific comments:

1. More detailed descriptions concerning generating knock-in (KI) of AID (JAZ)-mVenus-PGK-puro in PRDM14 locus by CRISPR-Cas9 are helpful to follow this KI strategy.

A detailed description of the KI generation is provided in Materials and Methods. We have also added a brief KI strategy description to the legend of Fig.3 in the revised manuscript to accompany the schematic in Fig.3A.

First, the cells were electroporated or lipofected with two plasmids: 1) gRNA sequence targeting the vicinity of the stop codon (of *PRDM14* or *SOX17*) ligated into BbsI-digested pX330 WT Cas9 construct (Cong et al. 2013); 2) HDR (homology-directed repair) donor template (Fig.1A) containing the sequences to be appended by KI, flanked by two 800-bp homology arms; additionally MC1-DTA was added after the 3' homology arm to reduce random integration of the vector (Kobayashi et al. 2017). The knocked-in sequences were: AID (*AtAID*⁴⁴) or T2A or JAZ (*OsJAZ*⁴³) tag tag followed by Venus fluorescent protein and a positive/negative PGK-puromycin- Δ TK selection cassette flanked by Rox sites to allow optional removal by transient Dre expression (Kobayashi et al. 2017).

Transfected cells were selected with puromycin, the resulting clones were picked (Kobayashi et al. 2017) and genotyped; homozygous lines were expanded. Genotyping amplicons were sequenced by Sanger sequencing to confirm correct targeting. In the case of AID and T2A lines, the selection cassette was removed by transient expression of Dre recombinase, followed by negative selection using FIAU. Correct homozygous clones were then transfected with PiggyBac constructs encoding the corresponding hormone receptor (TIR1 for AID and COI1B for JAZ) under the control of the CAG promoter followed by an IRES and a hygromycin resistance gene (Fig.3A). After hygromycin selection clones were picked and genotyped; the homogeneity of myc-TIR1 or HA-COI1B expression was checked by IF (using antibodies against epitope tags used). The depletion of target proteins upon hormone treatment was confirmed by IF and flow cytometry.

2. Fig. 5A, page 13: It is unclear why stabilization of PRDM14 at day 1 was not so effective as that at day 0 to rescue decreased PGCLC by IAA.

Similar to other overexpression rescue experiments, ProteoTuner system relies on transgene expression, which might result in heterogeneous expression levels. While we chose a clone with near-endogenous PRDM14 levels, a possibility for altered specification kinetics exists. To investigate this possibility, we performed a time-course rescue using the double degron (PRDM14-AID/TIR1-JAZ degron, see Fig.5 in the revised manuscript) to restore endogenous PRDM14 protein pool. Consequently, an earlier PRDM14 protein restoration led to more efficient hPGCLC specification rescue (Rebuttal Figure 2 and Fig.5E in the revised manuscript). For D0 Cor-mediated rescue, we obtained similar results as previously (Fig.5D) with significant restoration of hPGCLC specification efficiency in the presence of both IAA and Cor. The addition of Cor on D1-2 led to a slight, but insignificant, increase in hPGCLC specification.

Overall, this leads us to conclude that while PRDM14 expression level is low to moderate in D0-D1 nascent hPGCLCs (Fig.2A), it is nevertheless crucial at these early stages for hPGCLC specification. This outcome is consistent with the observation that earlier PRDM14 depletion by IAA (from D0 or D1) results in a stronger phenotype than depletion from D2 (Fig.4D).

Note also that that hPGCLC specification is likely to be an asynchronous process, with some cells responding earlier than others with respect to the upregulation of key germ cell markers, as is evident from relative expression dynamics of SOX17, BLIMP1, PRDM14, OCT4 and KLF4 (Rebuttal Figures 4, 5 and 7). It is possible that PRDM14 overexpression in cells that start upregulating SOX17 and BLIMP1 enhances their propensity to adopt hPGCLC fate by accelerating the expression of PRDM14 target genes and consolidating germ cell identity.

Rebuttal Figure 2. Selection of JAZ degron sequence and its validation in hESCs. Time-course of desensitisation to auxin via the AID/JAZ degron switch. Note that hPGCLC induction efficiency is only significantly rescued if Cor is supplemented from D0. Addition of Cor alone (without IAA) does not alter hPGCLC specification efficiency. Data show mean \pm -SD for n=4 (2 inductions of 2 clones), ns – not significant, **** P<0.0001 (Two-way ANOVA followed by Sidak’s multiple comparison test).

3. Fig. 6G, Suppl. Fig. 5G, Suppl. Fig. 6B: GO analysis for down-regulated genes by PRDM14 depletion in hPGCLCs may be informative.

We have now included the GO analysis for downregulated genes in hPGCLCs in the same figure (Fig.6G in the revised manuscript, also shown here as Rebuttal Figure 3). Top terms here included protein biosynthesis-related processes, negative regulation of WNT signalling and somatic stem cell population maintenance. Notably, the genes in the latter term included *NANOG* and *LIN28A*, TFs that are required for PGC development in mice (Yamaguchi et al. 2009; Chambers et al. 2007; West et al. 2009).

Rebuttal Figure 3. Gene ontology (GO) analysis on downregulated genes in PRDM14-deficient hPGCLCs. Top non-redundant GO terms are shown as $-\log_{10}(p\text{-value})$. For complete list of GO terms see Supplementary Table 10.

4. Page 26, the last two lines, ‘after the upregulation of SOX17 and BLIMP1 at the onset of hPGCLC specification (Fig.2 and Suppl. Fig.2)’: Those figures do not show the expression of BLIMP1, but Suppl. Fig.2 show the expression of TFAP2C.

Relative expression dynamics of SOX17 and BLIMP1, as well as of BLIMP1 and PRDM14 have been addressed previously (Irie et al. 2015). Nevertheless, we have performed a time-course IF analysis on this cell line for BLIMP1, PRDM14-Venus and OCT4 (Rebuttal Figure 5 and Suppl. Fig.2B in the revised manuscript). We have also repeated SOX17/PRDM14-Venus time-course adding OCT4 as another marker (Rebuttal Figure 4 and Suppl. Fig.2A in the revised manuscript). We note, however, that OCT4 is not hPGCLC-specific at the early stages of differentiation and

marks the majority of the EB at 12hrs - D1, and many SOX17/BLIMP1-negative cells on D2 (Rebuttal Figures 4-5 and (Irie et al. 2015)).

Consistent with our previous conclusions, we observed that PRDM14-Venus is upregulated after SOX17 and BLIMP1 (most SOX17⁺ or BLIMP1⁺ cells at 12 hrs are PRDM14-negative) with an increase in BLIMP1⁺PRDM14⁺ or SOX17⁺PRDM14⁺ cells from D1 onwards (Rebuttal Figures 4A-B and 5A-B). Note that many of the remaining BLIMP1⁺PRDM14⁻ or SOX17⁺PRDM14⁻ cells might belong to other lineages, as they lack OCT4 expression (Rebuttal Figures 4A and 5A). Indeed, PRDM14 marks the majority of BLIMP1⁺OCT4⁺ / SOX17⁺OCT4⁺ cells in D3-D5 embryoid bodies (Rebuttal Figures 4C and 5C).

Importantly, on D2, a higher proportion of BLIMP1⁺ rather than SOX17⁺ cells express PRDM14 (Rebuttal Figures 4B and 5B), which agrees with the established order of SOX17 expression followed by BLIMP1 during hPGCLC specification. The majority of BLIMP1⁺ cells are likely to be SOX17⁺ and are therefore expected to show upregulation of PRDM14. By contrast, some SOX17⁺ cells destined for the hPGCLC fate might be at an earlier stage and lack BLIMP1 expression and thus unlikely to have advanced far enough to express PRDM14. Thus BLIMP1⁺ cells compared to SOX17⁺ cells have a greater propensity to be PRDM14⁺

Rebuttal Figure 4. PRDM14 is upregulated after SOX17 during hPGCLCs induction. (A) Time-course IF analysis showing SOX17, PRDM14-Venus and OCT4 expression in embryoid body (EB) sections throughout hPGCLC specification from PRDM14-AID-Venus fusion reporter cell line. hPGCLCs were induced by cytokines and EBs were collected at 12hrs, and on D1-D5. Representative images for all timepoints are shown. Dashed line highlights SOX17⁺ cells. Nuclei were counterstained by DAPI. Scale bar is 100 μ m. (B) Quantification of results from (A) showing the percentage of PRDM14-Venus⁺SOX17⁺ cells as mean \pm -SD of n=11 EB sections from 2 hPGCLC inductions. (C) Quantification of results from (A) showing the percentage of PRDM14-Venus⁺SOX17⁺OCT4⁺ cells as mean \pm -SD of n=3 EB sections from 1 hPGCLC induction.

B PRDM14 expression in BLIMP1⁺ cells

C PRDM14 expression in BLIMP1⁺OCT4⁺ cells

Rebuttal Figure 5. PRDM14 is upregulated after BLIMP1 during hPGCLCs induction. (A) Time-course IF analysis showing BLIMP1, PRDM14-Venus and OCT4 expression in embryoid body (EB) sections throughout hPGCLC specification from PRDM14-AID-Venus fusion reporter cell line. hPGCLCs were induced by cytokines and EBs were collected at 12hrs, and on D1-D5. Representative images for all timepoints are shown. Dashed line highlights BLIMP1⁺ cells. Nuclei were counterstained by DAPI. Scale bar is 100 μ m. **(B)** Quantification of results from (A) showing the percentage of PRDM14-Venus⁺BLIMP1⁺ cells as mean \pm -SD of n=3 EB sections from 1 hPGCLC induction. **(C)** Quantification of results from (A) showing the percentage of PRDM14-Venus⁺BLIMP1⁺OCT4⁺ cells as mean \pm -SD of n=3 EB sections from 1 hPGCLC induction.

5. Page 3, the last line: 100 μ M may be 100 μ m.

We apologise for the error and have corrected it in the revised version.

Reviewer #3 (Remarks to the Author):

Overall, this is a well-conceived study with interesting results that point to PRDM14's role in hPGCLC differentiation but with different targets. One of the major criticisms is that some of the findings are based on the analysis of what appears to be n=1 replicate or in some cases n=1 cell. The work would be better served by additional replicates to confirm key findings as indicated below.

We have considered the Reviewer's comments carefully and address them as outlined in detail below.

For Sup fig 1A the authors claim both nuclear & cytoplasmic staining for PRDM14 protein. It would be helpful to see OCT4 and SOX17 merged with PRDM14 to confirm the intracellular localization. Furthermore, the legend indicates that both samples are female, but the figure is labelled 8wk male and 7wk female. Which is correct? If the figure label is correct, this would imply that the 7 wk female has nuclear PRDM14 in the hPGCs and the 8 wk male has cytoplasmic PRDM14 staining. For the 9 wk female sorted hPGCs, PRDM14 was nuclear. Given this apparent dynamic and possibly sex-specific localization of PRDM14 samples, more replicates around these time points from both male and female embryos are needed before drawing conclusions regarding PRDM14 protein in hPGCs.

We regret that there was an error in the figure label – Suppl. Figure 1A did indeed show two female samples; this error has been corrected in the revised manuscript. We have also performed several more stainings on additional female and male gonadal sections, including OCT4 as a nuclear and VASA as a cytoplasmic hPGC marker (Suppl. Fig.1 in the revised manuscript and Rebuttal Figure 6). Some sections showed predominantly nuclear localisation of PRDM14, while other samples contained hPGCs with cytoplasmic (and nuclear) PRDM14 signal, ranging from ~15% to ~70% in different sections. None of the tested samples showed hPGCs with exclusively cytoplasmic localisation of PRDM14.

From the available data, it does not however seem that PRDM14 localisation is a consequence of hPGC stage or sex. Many more human fetal samples and stages as and when they become available will need to be tested with additional batches of PRDM14 antibodies in the future. Nevertheless, this study shows that PRDM14 has a role in hPGCLC specification, which was the primary objective of our study.

Rebuttal Figure 6. Nuclear PRDM14 localisation is observed in many human gonadal PGCs. Additional samples, not previously presented in the paper, are shown. (A) Representative IF images of PRDM14 staining in one female gonad (Wk9) and two male gonads (Wk7 and Wk8). hPGCs were marked by OCT4 (nuclear PGC marker) and VASA (cytoplasmic PGC marker). Note both nuclear and cytoplasmic PRDM14 in “Male Wk7” sample. Nuclei were counterstained by DAPI. Scale bar is 100 µm.

For Fig 2: It was unclear whether the lack of PRDM14 in SOX17 cells at 12 hr is because the SOX17 positive cells are endodermal (not germ cells) or as the authors suggest that PRDM14 is not required at this stage of hPGCLC formation. To address this the authors should stain with a second hPGC marker to determine whether the SOX17+ cells are indeed hPGCLCs at 12 hours. We see in the supplement there is also an image showing TFAP2C +/SOX2- PRDM14+ cell at 12hr, but a similar argument can also be made for the possibly the TFAP2C single positive cell is also not a hPGCLCs.

(Please note that the answer to this question partially overlaps with our response to question 4 of Reviewer #2).

Currently, there is no known single marker at 12hrs for the unequivocal detection of hPGCLCs. It is therefore possible that a small percentage of SOX17⁺ cells at 12hrs could belong to other lineages. However, virtually none of the SOX17-positive cells at 12hrs are PRDM14⁺, suggesting that potential nascent germ cells also lack PRDM14 at this stage. To test this hypothesis, as suggested by Reviewer #3, we co-stained Venus and SOX17 with other markers, namely OCT4, BLIMP1 and KLF4 (Rebuttal Figures 4, 5 and 7).

Note also that expression of BLIMP1 is not exclusive to hPGCLCs in mammals and could mark rare cells of other lineages, including extraembryonic tissues and mesoderm (Hopf, Viebahn, and Puschel 2011; Vincent et al. 2005). OCT4, while specific to germ cells from D3 onwards, was detected in most cells of the EB at 12hrs and on D1, as well as in many BLIMP1/SOX17⁻ cells on D2 (Rebuttal Figures 4 and 5; (Irie et al. 2015)). There were also many SOX17-positive cells that lacked OCT4 at 12hrs – D2 and it is difficult to conclude if these cells belong to other lineages or if OCT4 is upregulated in hPGCLCs after SOX17. Thus, there is currently a clear lack of a single germ-cell-specific marker at the onset of hPGCLC specification. Lineage tracing experiments would be helpful to address this interesting question in the future.

Consistent with our previous conclusions, we observe that PRDM14-Venus is upregulated after SOX17 and BLIMP1 (most SOX17⁺ or BLIMP1⁺ cells at 12hrs are PRDM14-negative); there is an increase in BLIMP1⁺PRDM14⁺ or SOX17⁺PRDM14⁺ cells from D1 onwards (Rebuttal Figures 4A-B and 5A-B). Note the remaining BLIMP1⁺PRDM14⁻ or SOX17⁺PRDM14⁻ cells might belong to other lineages, as they lack OCT4 expression (Rebuttal Figures 4A and 5A). Indeed, PRDM14 marks the majority of BLIMP1⁺OCT4⁺ / SOX17⁺OCT4⁺ cells in D3-D5 embryoid bodies (Rebuttal Figures 4C and 5C).

Importantly, on D2, a higher proportion of BLIMP1⁺ rather than SOX17⁺ cells express PRDM14 (Rebuttal Figures 4B and 5B), which agrees with the established order of SOX17 expression followed by BLIMP1 during hPGCLC specification. The majority of BLIMP1⁺ cells are likely to be SOX17⁺ and are therefore expected to show upregulation of PRDM14. By contrast, some SOX17⁺ cells destined for the hPGCLC fate might be at an earlier stage and lack BLIMP1 expression and thus unlikely to have advanced far enough to express PRDM14. Accordingly, BLIMP⁺ cells have a greater propensity to be PRDM14⁺, compared to SOX17⁺ cells.

We decided to evaluate other potential hPGCLC markers and performed staining for KLF4, which is reported to be specifically expressed in hPGCs (Tang et al. 2015), but repressed in mouse germ cells (Kurimoto et al. 2008). Notably, KLF4 was not expressed in hPGCLC until D3; KLF4 expression followed that of PRDM14 (Rebuttal Figure 7 and Suppl. Fig.2G in the revised manuscript). qPCR analysis also showed downregulation of *KLF4* in PRDM14-depleted hPGCLCs (Fig.6E), suggesting that PRDM14 might be upstream of *KLF4* in hPGCLCs. This could suggest that PRDM14 is important for shaping and consolidating the unique human hPGC TF network.

Rebuttal Figure 7. KLF4 is upregulated after PRDM14 during hPGCLCs induction. Time-course IF analysis showing SOX17, PRDM14-Venus and KLF4 expression in embryoid body (EB) sections throughout hPGCLC specification from PRDM14-AID-Venus fusion reporter cell line. hPGCLCs were induced by cytokines and EBs were collected at 12hrs, and on D1-D5. Representative images for all timepoints are shown. Dashed line highlights SOX17⁺ cells. Note KLF4 upregulation from D3 of differentiation. Nuclei were counterstained by DAPI. Scale bar is 100 μm.

Please label the images with +Cyto (as was done for no cyto) to make the interpretation clearer.

The relevant images in Suppl. Fig.2E-F in the revised manuscript have been labelled “+cytokine” or “no cytokine” to denote the presence or absence of cytokines in the medium, respectively.

For Fig4. B&C. The authors have indicated based on previous studies that PRDM14 is required for pluripotent self-renewal, therefore inducing a knockout in PRDM14 at D0, followed by a loss of hPGCLCs may be due to either a direct role for PRDM14 in hPGCLC specification, or an indirect role related to the general loss of pluripotency and propensity for somatic cell differentiation. Given uncoupling this result (direct verse indirect) is central to determining the role of PRDM14 in hPGCLC specification, the authors need to show more specifically in the earliest stages of PRDM14KO that either the somatic cell differentiation is unaffected, or if it is affected, then whether this does or does not have an impact on hPGCLC specification. Particularly, given that removing PRDM14 in specified hPGCLCs at day 4 has not affect on hPGCLCs.

Indeed, PRDM14 is a critical regulator of pluripotency in hESCs: it promotes the expression of the pluripotency-associated genes and prevents upregulation of differentiation-associated transcripts (Chia et al. 2010). We further discuss the role of PRDM14 in human pluripotency regulation in our reply to the next question (see below).

It is unlikely that PRDM14 depletion causes the observed hPGCLC phenotype by interfering with pluripotency regulation, since PRDM14 is transiently downregulated during the transition from pluripotency to hPGCLC differentiation (Fig.2). Furthermore, PRDM14 depletion from D1 or D2 still causes a significant decrease in hPGCLC specification (Fig.4D) and at these stages all cells in the EBs have likely exited from pluripotency as judged by the loss of SOX2 expression (Suppl. Fig.2F in the revised manuscript).

Finally, to distinguish between direct and indirect effect of PRDM14 depletion of hPGCLC specification, we have tested PRDM14 depletion for 1 passage in pluripotent PGC-competent cells, allowing PRDM14 restoration only at the onset of differentiation (Suppl. Fig5H-I in the revised manuscript). This did not lead to a reduction in hPGCLC specification efficiency (Suppl. Fig5I in the revised manuscript), indicating a specific role in hPGCLC specification as opposed to competence. Altogether, these results suggest that PRDM14 depletion causes a decrease in hPGCLC specification directly and not by interfering with pluripotency.

We agree that soma differentiation might be affected by PRDM14 depletion at the onset of hPGCLC induction. We anticipate however the soma composition to be heterogeneous, especially as our protocol is optimised for germ cell fate. To address this, we performed qPCR analysis, comparing expression of lineage markers in D2 hPGCLCs and soma (Rebuttal Figure 8A). D2 was chosen as it is the earliest time point when hPGCLCs can be sorted using specific expression of NANOS3-tdTomato in our reporter cell line. This showed high expression of primitive endoderm markers *GATA6*, *PDGFRA* and *SOX7*, as well as trophoctoderm marker *HAND1* in somatic cells compared to hPGCLCs. Of note, we could not detect *FOXA2* (neither in hPGCLCs nor in soma), and *SOX17* expression in soma was very low, consistent with its predominant expression in hPGCLCs. This suggests that EB soma does not contain definitive endoderm-like cells. IF analyses also showed that EBs lack SOX2 expression (Suppl. Fig.2F in the revised manuscript), indicating exit from the pluripotent state. Importantly, PRDM14 depletion resulted in a slight increase in *PDGFRA* levels, while all other assessed markers remained unchanged in soma (Rebuttal Figure 8A).

We went on to examine D4 soma composition by IF and confirmed that many soma cells express GATA6 (Rebuttal Figure 8B). Since the majority of soma cells are SOX17-negative, GATA6 expression might be indicative of primitive endoderm and not definitive endoderm lineage. Indeed, previous studies have shown that SOX17 is dispensable for initial primitive endoderm commitment (Kanai-Azuma et al. 2002; Shimoda et al. 2007). Consistent with D2 qPCR results (Rebuttal Figure 8A), the percentage of GATA6⁺ cells remained unchanged in PRDM14-deficient EBs (Rebuttal Figure 8B,D). However, we noticed a slight increase (from 6.7% to 14%) in the proportion of HNF4A⁺ somatic cells in EBs lacking PRDM14 (Rebuttal Figure 8C,E).

Altogether, these results suggest that there is only a minor increase in proportion of primitive endoderm-like cells in soma of PRDM14-deficient EBs. It is unlikely that this would cause significant changes in hPGCLC specification as observed in our study. Therefore, we posit that PRDM14 loss affects hPGCLC specification directly, in line with its regulation of many PGC-related genes (Fig.6B).

In the text and in Sup Fig 4, the authors state that PRDM14 depletion in 4i for one passage had no effect on hPGCLC specification. If PRDM14 is essential for human pluripotent stem cell self renewal, how did the cells remain pluripotent without PRDM14? Is PRDM14 not needed when cells are maintained in 4i conditions verses regular primed?

We thank the Reviewer for these relevant questions. The seminal paper by Chia and colleagues (Chia et al. 2010) described the importance of PRDM14 for human pluripotency, although the transcriptional changes were often less pronounced than in our study; we attribute the differences to the use of knockdown versus AID. However, Fig.6C shows that the overall changes were similar (most upregulated and downregulated genes in our study were also reported by Chia et al). This suggests that PRDM14 plays essentially similar roles in conventional (Chia et al 2010) and PGC-competent hESC (this study). Indeed, we observed a decrease in expression of pluripotency-related genes and an increase in differentiation-related markers (Fig.6A, Fig.6F, Suppl. Fig.6E in the revised manuscript).

While we did not carry out detailed studies on the impact of PRDM14 depletion on 4i, we considered it important for this study to test if the depletion of PRDM14 in 4i cells has an effect on subsequent hPGCLC specification; we found no detectable effect on hPGCLC specification (Suppl. Fig.5I in the revised manuscript). Had this not been the case, we would have been prompted to examine the phenotype in more detail. Nevertheless, it will be of interest in the future to examine the role of PRDM14 on many diverse pluripotent stem cell types, including PGC-competent (4i), primed and naïve hESCs. The reagents we have developed will be of value in such studies; however, this is outside the scope of our study, where the focus is on hPGCLC specification.

In Fig 5. D the authors performed a rescue with a shield which was successful on D0, but not successful at D1. Given this timing, what are the authors claiming? I am confused.

We kindly refer the Reviewer to our response to the comment #2 from Reviewer 2 (see above).

The authors performed RNA-Seq and show in Figure 6 NANOG down regulation. However in Figure 3 NANOG protein is still expressed in PRDM14 depleted cells. Which is correct?

RNA-seq (Fig.6) was performed after 3 days of IAA treatment, while IF (Fig.3B) was performed after only 25 minutes of PRDM14 depletion, which was presumably insufficient time to affect NANOG abundance at the protein level.

Rebuttal Figure 8. Soma is predominantly composed of primitive endoderm-like cells in D2 and D4 EBs. (A) qPCR on hPGCLCs and soma from D2 EBs induced with or without IAA from PRDM14-AID hESCs (c111 and c121) and the parental control (no TIR1). hPGCLCs were sorted as NANOS3-tdTomato⁺AP⁺ cells, while soma denotes the double-negative population from the same experiments. Data show gene expression relative to no IAA hPGCLCs and normalized to GAPDH, mean \pm SEM of n=2 for PRDM14-AID (2 clones) or n=1 for no TIR1 control; * P<0.05, ** P<0.01 (multiple t-tests). (B-C) Representative images showing IF analysis of GATA6 (B) or HNF4A (C) expression D4 EBs induced with or without IAA from PRDM14-AID hESCs. OCT4 marks hPGCLCs. Scale bar is 100 μ m. (D) Quantification of results from (B), including staining in no TIR1 control line. (E) Quantification of results from (C), including staining in no TIR1 control line.

Furthermore, analysis of the 2015 Sasaki et al., Cell Stem Cell paper on hPGCLC differentiation from primed hESCs shows that hPGCLCs have background levels of PRDM14 RNA. How can these different RNA-seqs be reconciled? Is Prdm14 RNA uniquely regulated in human pluripotent stem cells and the earliest stages of human germ cell differentiation thus leading to different results on different platforms? The expression of Prdm14 RNA needs to be better described.

It has been shown, by us and others, that the PRDM14 levels are lower in hPGCLCs and hPGCs compared to hESCs; the expression is specific, albeit of moderate levels in germ cells (Sasaki et al. 2015; Irie et al. 2015). The paper mentioned by Reviewer #3 states “modest/low levels of *T* and *PRDM14*”, indicating that PRDM14 transcript is present in the hPGCLCs induced using a distinct protocol from different cell lines (Sasaki et al. 2015). To support this view, we provide a heatmap (Rebuttal Figure 9) showing the expression of PRDM family members in hESCs, gonadal hPGCs and soma, as well as in hPGCLCs obtained from pre-mesendoderm (Kobayashi et al. 2017), in a two-step system that resembles iMeLC reported independently (Sasaki et al. 2015). Notably, PRDM1 (BLIMP1) and PRDM14 are the only two PRDM family members that are absent in the neighbouring soma but expressed in hPGCs (Rebuttal Figure 9). Thus, irrespective of the method used for the competent state induction, PRDM14 is moderately and specifically expressed in hPGCLCs.

[Redacted]

Rebuttal Figure 9. Heatmap showing PRDM family gene expression in hESCs, germ cells and soma. RNA-seq data is from (Irie et al. 2015) (hPGCs and soma) or unpublished (other samples). Note specific PRDM1 (BLIMP1) and PRDM14 expression in hPGCs and hPGCLCs. PRDM14 is also expressed in hESCs, at a higher level than in germ cells.

It appears that one of the P14-AID lines in Figure 6 is quite different in the PCA. Given that these are not true biological replicates but rather N=1 for two different cell lines it will be important to repeat these experiments from at least 1-2 more replicates from each cell line.

The two IAA-sensitive clones were obtained from the same parental line by PiggyBac-delivered TIR1 construct. Therefore, there might be a minimal change in the genetic background due to multiple PiggyBac integrations. We reasoned that using two clones is a more stringent approach than repeating the experiment with the same line (as was done for the parental control). Crucially, despite these being two independent clones, they still cluster together on both the PCA plot (Fig.6D) and by using unsupervised hierarchical clustering (Suppl. Fig.6A in the revised manuscript). Furthermore, the top changes in gene expression were independently confirmed by qPCR in several cell lines using independent hPGCLC inductions (Fig.6E and Suppl. Fig.6C-D in the revised manuscript).

In the discussion, the authors claim that PRDM14 is likely downstream of SOX2, however, this would be contradictory to the results of the RNA seq, as a KO of PRDM14 in hESCs leads to increased SOX2 expression. Are the authors suggesting that PRDM14 is a negative regulator?

We are grateful to the Reviewer for pointing this out. Multiple examples can be found in the literature for reciprocal transcription factor regulation (Chew et al. 2005; Niwa 2018). It is therefore possible that PRDM14 and SOX2 are involved in their mutual expression regulation. RNA-seq did not detect significant changes in *SOX2* expression in hESCs upon PRDM14 depletion for 3 days, and qPCR detected only a small increase. Further culture (for 2 more passages) without PRDM14 led to upregulation of markers of neuronal differentiation, including *SOX2* (Suppl. Fig 6E in the revised manuscript). It is therefore likely that this increase in *SOX2* expression is not a direct consequence of PRDM14 depletion.

Our statement on SOX2 regulating PRDM14 comes from previous ChIP-seq and expression studies strongly suggesting that PRDM14 expression is directly regulated by SOX2 in hESCs (Boyer et al. 2005; Wang et al. 2012). We therefore posit that at the very onset of hPGCLC specification when PRDM14 is still controlled by SOX2, the abrupt SOX2 repression leads to a transient PRDM14 downregulation (Fig.2); PRDM14 is then upregulated in a germ-cell specific manner by other regulators, presumably SOX17 and BLIMP1 either directly or indirectly. As in other instances, regulation of transcription factors is context-dependent.

Next, the hypothesis is that PRDM14 is upregulated after expression of SOX17 and BLIMP 1 (ref fig 2, sup fig 2), but BLIMP was never evaluated so how can this claim be made?

We kindly refer the Reviewer to our response to the comment #4 from Reviewer 2 (see above).

The claim that PRDM14 potentially plays a role in UHRF1 in hPGCLCs, similar to mice, would be strengthened if they quantified how often Ki67+ UHRF+ SOX17+ cells occurred. Not convinced by showing one cell that fits this criterion.

In addition, the authors discuss PRDM14-def hPGCLCs as exhibiting de-repressed UHRF1, but there is just one cell in an IF image to prove this, and given the importance of this potential result, stronger evidence is needed to support this claim.

We apologise for not providing quantification in the original version of the paper. UHRF1 retention in proliferating hPGCLCs was indeed rare and we have now performed additional stainings to verify these findings. To exclude the possibility that UHRF1-retaining SOX17-positive cells were rare endoderm-like cells and not hPGCLCs, we performed IF analysis replacing SOX17 with OCT4 as a more stringent hPGCLC marker at this stage. In this experiment we also observed some UHRF1-retaining proliferative hPGCLCs (Rebuttal Figure 10 and Suppl. Fig.8C,G in the revised manuscript).

In addition to some cells derepressing UHRF1 very significantly to levels similar to those in soma, many cells also displayed intermediate levels of UHRF1 derepression (Rebuttal Figure 10A). To quantify this, we measured mean fluorescence intensities (MFI) of all soma cells and proliferating (Ki67-positive) germ cells. Initially, this showed an increase in MFI of UHRF1 in PRDM14-depleted hPGCLCs. However, there was also an increase in UHRF1 MFI in soma of these samples. To avoid potential exposure artefacts between slides, we normalised fluorescent intensities of PRDM14-AID “+IAA” samples by the ratio of median MFIs of “soma +IAA” to “soma no IAA”. While the trend towards higher UHRF1 fluorescence intensity in PRDM14-depleted proliferating hPGCLCs was preserved, it was no longer statistically significant (Rebuttal Figure 10B). Using these normalised MFI values, we counted the numbers of proliferating (Ki67-positive) hPGCLCs (OCT4-positive) that show UHRF1 MFI higher than 1000 and found their percentage to increase after IAA treatment (Rebuttal Figure 10C).

Overall these results point to a potential derepression of UHRF1 in hPGCLCs lacking PRDM14, but the effect is heterogeneous, potentially due to the early stage of hPGCLC differentiation. It is possible that UHRF1 derepression would become more pronounced in PRDM14-deficient hPGCLCs at later stages. It is however difficult to address this question comprehensively in this *in vitro* system.

Another statement in the discussion is the claim that PRDM14 mutant hPGCLCs do not de-repress EHMT2, but there was no evidence for this statement in this paper, the first time they mention EHMT2 is the discussion, there is no H3K9me2 staining.

We performed similar IF analyses as for UHRF1 (see above) to assess H3K9me2 levels in hPGCLCs induced with or without PRDM14 (Rebuttal Figure 11 and Suppl. Fig.8D,F,H in the revised manuscript). In agreement with the global depletion of H3K9me2 observed in mouse and human PGCs (Seki et al. 2005; Tang et al. 2015), H3K9me2 levels were lower in hPGCLCs (induced with or without IAA) compared to the neighbouring soma (Rebuttal Figure 11A). Initial quantification revealed higher H3K9me2 MFI in proliferating hPGCLCs+IAA than in control (no IAA), but this change was not significant after normalisation of background fluorescence (performed as for UHRF1, see above; Rebuttal Figure 11B,C)

Thus, there is no evidence for H3K9me2 retention in PRDM14-depleted proliferating D4 hPGCLCs compared to controls. It is possible, however, that the change could manifest itself at later developmental stages, which cannot be assessed using this *in vitro* system.

Rebuttal Figure 10. PRDM14 depletion results in UHRF1 derepression in a subset of proliferating hPGCLCs. (A) IF analysis of UHRF1 expression in D4 PRDM14-AID EBs induced in the presence of auxin. hPGCLCs are marked by OCT4, proliferating cells are marked by Ki-67. OCT4-positive cells are highlighted by a dashed line. White arrows show examples of PRDM14-deficient proliferating hPGCLC that retain UHRF1. Scale bar is 100 μ m. (B) UHRF1 mean fluorescence intensity in D4 EBs. UHRF1 fluorescence intensities in hPGCLCs+IAA and soma+IAA were normalised by the ratio of median fluorescence in “soma+IAA” to “soma no IAA”. (C) Quantification of UHRF1-positive cells in proliferating hPGCLCs and soma of PRDM14-AID and control EBs. Normalised fluorescence values were used. Numbers of UHRF1 positive and total cells counted are shown for each sample.

Rebuttal Figure 11. PRDM14 depletion does not alter H3K9me2 levels in D4 hPGCLCs. (A) IF analysis of H3K9me2 levels in D4 PRDM14-AID EBs. hPGCLCs are marked by OCT4, proliferating cells are marked by Ki-67. OCT4-positive cells are highlighted by a dashed line. Scale bar is 100 μ m. (B) H3K9me2 mean fluorescence intensity in D4 EBs. H3K9me2 fluorescence intensities in hPGCLCs+IAA and soma+IAA were normalised by the ratio of median fluorescence in “soma+IAA” to “soma no IAA”. (C) Quantification of H3K9me2-positive cells in proliferating hPGCLCs and soma of PRDM14-AID EBs. Normalised fluorescence values were used. Numbers of H3K9me2 positive and total cells counted are shown for each sample.

References

- Boyer, L. A., T. I. Lee, M. F. Cole, S. E. Johnstone, S. S. Levine, J. P. Zucker, M. G. Guenther, R. M. Kumar, H. L. Murray, R. G. Jenner, D. K. Gifford, D. A. Melton, R. Jaenisch, and R. A. Young. 2005. 'Core transcriptional regulatory circuitry in human embryonic stem cells', *Cell*, 122: 947-56.
- Brosh, R., I. Hrynyk, J. Shen, A. Waghray, N. Zheng, and I. R. Lemischka. 2016. 'A dual molecular analogue tuner for dissecting protein function in mammalian cells', *Nat Commun*, 7: 11742.
- Chambers, I., J. Silva, D. Colby, J. Nichols, B. Nijmeijer, M. Robertson, J. Vrana, K. Jones, L. Grotewold, and A. Smith. 2007. 'Nanog safeguards pluripotency and mediates germline development', *Nature*, 450: 1230-4.
- Chew, J. L., Y. H. Loh, W. Zhang, X. Chen, W. L. Tam, L. S. Yeap, P. Li, Y. S. Ang, B. Lim, P. Robson, and H. H. Ng. 2005. 'Reciprocal transcriptional regulation of Pou5f1 and Sox2 via the Oct4/Sox2 complex in embryonic stem cells', *Mol Cell Biol*, 25: 6031-46.
- Chia, N. Y., Y. S. Chan, B. Feng, X. Lu, Y. L. Orlov, D. Moreau, P. Kumar, L. Yang, J. Jiang, M. S. Lau, M. Huss, B. S. Soh, P. Kraus, P. Li, T. Lufkin, B. Lim, N. D. Clarke, F. Bard, and H. H. Ng. 2010. 'A genome-wide RNAi screen reveals determinants of human embryonic stem cell identity', *Nature*, 468: 316-20.
- Cong, L., F. A. Ran, D. Cox, S. Lin, R. Barretto, N. Habib, P. D. Hsu, X. Wu, W. Jiang, L. A. Marraffini, and F. Zhang. 2013. 'Multiplex genome engineering using CRISPR/Cas systems', *Science*, 339: 819-23.
- Hopf, C., C. Viebahn, and B. Puschel. 2011. 'BMP signals and the transcriptional repressor BLIMP1 during germline segregation in the mammalian embryo', *Dev Genes Evol*, 221: 209-23.
- Irie, N., L. Weinberger, W. W. Tang, T. Kobayashi, S. Viukov, Y. S. Manor, S. Dietmann, J. H. Hanna, and M. A. Surani. 2015. 'SOX17 is a critical specifier of human primordial germ cell fate', *Cell*, 160: 253-68.
- Kanai-Azuma, M., Y. Kanai, J. M. Gad, Y. Tajima, C. Taya, M. Kurohmaru, Y. Sanai, H. Yonekawa, K. Yazaki, P. P. Tam, and Y. Hayashi. 2002. 'Depletion of definitive gut endoderm in Sox17-null mutant mice', *Development*, 129: 2367-79.
- Kobayashi, T., H. Zhang, W. W. C. Tang, N. Irie, S. Withey, D. Klisch, A. Sybirna, S. Dietmann, D. A. Contreras, R. Webb, C. Allegrucci, R. Alberio, and M. A. Surani. 2017. 'Principles of early human development and germ cell program from conserved model systems', *Nature*, 546: 416-20.
- Kurimoto, K., Y. Yabuta, Y. Ohinata, M. Shigeta, K. Yamanaka, and M. Saitou. 2008. 'Complex genome-wide transcription dynamics orchestrated by Blimp1 for the specification of the germ cell lineage in mice', *Genes Dev*, 22: 1617-35.
- Niwa, H. 2018. 'The principles that govern transcription factor network functions in stem cells', *Development*, 145.
- Sasaki, K., S. Yokobayashi, T. Nakamura, I. Okamoto, Y. Yabuta, K. Kurimoto, H. Ohta, Y. Moritoki, C. Iwatani, H. Tsuchiya, S. Nakamura, K. Sekiguchi, T. Sakuma, T. Yamamoto, T. Mori, K. Woltjen, M. Nakagawa, T. Yamamoto, K. Takahashi, S. Yamanaka, and M. Saitou. 2015. 'Robust In Vitro Induction of Human Germ Cell Fate from Pluripotent Stem Cells', *CELL STEM CELL*.
- Seki, Y., K. Hayashi, K. Itoh, M. Mizugaki, M. Saitou, and Y. Matsui. 2005. 'Extensive and orderly reprogramming of genome-wide chromatin modifications associated with specification and early development of germ cells in mice', *Developmental Biology*, 278: 440-58.
- Shimoda, M., M. Kanai-Azuma, K. Hara, S. Miyazaki, Y. Kanai, M. Monden, and J. Miyazaki. 2007. 'Sox17 plays a substantial role in late-stage differentiation of the extraembryonic endoderm in vitro', *J Cell Sci*, 120: 3859-69.
- Tang, W. W., S. Dietmann, N. Irie, H. G. Leitch, V. I. Floros, C. R. Bradshaw, J. A. Hackett, P. F. Chinnery, and M. A. Surani. 2015. 'A Unique Gene Regulatory Network Resets the Human Germline Epigenome for Development', *Cell*, 161: 1453-67.
- Vincent, S. D., N. R. Dunn, R. Sciammas, M. Shapiro-Shalef, M. M. Davis, K. Calame, E. K. Bikoff, and E. J. Robertson. 2005. 'The zinc finger transcriptional repressor Blimp1/Prdm1 is dispensable for early axis formation but is required for specification of primordial germ cells in the mouse', *Development*, 132: 1315-25.
- Wang, Z., E. Oron, B. Nelson, S. Razis, and N. Ivanova. 2012. 'Distinct lineage specification roles for NANOG, OCT4, and SOX2 in human embryonic stem cells', *CELL STEM CELL*, 10: 440-54.
- West, J. A., S. R. Viswanathan, A. Yabuuchi, K. Cunniff, A. Takeuchi, I. H. Park, J. E. Sero, H. Zhu, A. Perez-Atayde, A. L. Frazier, M. A. Surani, and G. Q. Daley. 2009. 'A role for Lin28 in primordial germ-cell development and germ-cell malignancy', *Nature*, 460: 909-U151.
- Yamaguchi, S., K. Kurimoto, Y. Yabuta, H. Sasaki, N. Nakatsuji, M. Saitou, and T. Tada. 2009. 'Conditional knockdown of Nanog induces apoptotic cell death in mouse migrating primordial germ cells', *Development*, 136: 4011-20.

REVIEWERS' COMMENTS:

Reviewer #1 (Remarks to the Author):

The authors adequately responded to my comments by providing sufficient technical information. I do not see any problems at all. Many congratulations!

Reviewer #2 (Remarks to the Author):

All of my concerns for the original manuscript have been properly addressed in the revised version.

Reviewer #3 (Remarks to the Author):

The authors have carefully considered all of my comments and I am satisfied with the new experiments and/or clarifications. The only change I recommend would be on lines 242-244 the authors discuss data not shown, either show the data (which is interesting as it relates to the function of PRDM14 in hPGCLCs verses endoderm) or delete the sentence referring to data we can't evaluate

Response to Reviewers on the manuscript: “**A critical role of PRDM14 in human primordial germ cell fate revealed by inducible degrons**” by Sybirna *et al.*

Responses appear in red font.

REVIEWERS' COMMENTS:

Reviewer #1 (Remarks to the Author):

The authors adequately responded to my comments by providing sufficient technical information. I do not see any problems at all. Many congratulations!

Reviewer #2 (Remarks to the Author):

All of my concerns for the original manuscript have been properly addressed in the revised version.

Reviewer #3 (Remarks to the Author):

The authors have carefully considered all of my comments and I am satisfied with the new experiments and/or clarifications. The only change I recommend would be on lines 242-244 the authors discuss data not shown, either show the data (which is interesting as it relates to the function of PRDM14 in hPGCLCs verses endoderm) or delete the sentence referring to data we can't evaluate

We are grateful to all Reviewers for the positive reception of the revised manuscript.

The sentence mentioned by Reviewer #3 has been deleted.